# Identifying Invariant Physical Dynamics Across Multiple Environments

## Abstract

Data-driven machine learning methods have been widely employed in dynamical systems, but they fail to generalize to unseen dynamical environments where data is sparse and noisy. While few recent works have proposed to address this problem by introducing domain adaptation techniques based on deep neural networks (DNNs), these black-box methods fail to explain and understand the underlying system behaviors. In this work, we propose an Invariant PhysicAl Dynamics identification framework (IPAD), designed to extract common physical laws from data collected from multiple environments. Specifically, IPAD combines Monte Carlo Tree Search (MCTS) and multi-environment reward to effectively uncover physical dynamics from imbalanced data across multiple environments. Moreover, it incorporates a variational formulation (VF) loss function to enhance the robustness in noisy conditions and then introduces a post-hoc purification approach to further refine discovered equations. We also theoretically prove the convergence rate of VF for symbolic regression. The evaluation results demonstrate that IPAD significantly outperforms existing methods in discovering invariant physical dynamics across both simulated and real-world datasets. The source code will be publicly available upon publication.

## 1 Introduction

Data-driven machine learning (ML) models have been widely employed in dynamical systems, but they are prone to learning spurious correlations, making them hard to adapt to unseen dynamic environments. In reality, dynamical systems are often governed by the underlying physical laws. Due to different external forces and environmental conditions, the underlying dynamics may comply with the same physical laws but have different system parameters. For instance, a car's acceleration follows Newton's second law of motion, but the friction force may vary across different road conditions under different environments. Thus, it is crucial to learn invariant physical laws from observed data across multiple environments to enhance the generalization capability of ML for dynamical systems.

To achieve this, recent studies have developed different domain adaptation techniques to improve the generalization ability of learning the dynamics of physical systems. For instance, Kirchmeyer et al. (2022) introduced CoDA, a context-informed dynamics adaptation framework for fast adaptation to unknown domains. Additionally, a recent study Huang et al. (2023) proposed GG-ODE, an approach for learning multi-agent system dynamics across various environments. While existing methods can generalize to unknown domains using black-box deep neural networks (DNNs), they fail to explicitly uncover the invariant physical laws across multiple environments. As a result, it is very hard for us to understand the system behaviors. Following prior work Huang et al. (2023), our hypothesis is that the underlying system dynamics are commonly governed by the same (or only in part) physical dynamics but with different parameters. Hence, the goal of this work is to identify *interpretable invariant physical laws from multiple environments* where data is scarce, noisy, and imbalanced.

To reach this goal, we introduce an **I**nvariant **P**hysic**A**l **D**ynamics identification framework, named IPAD, that uncovers invariant physical laws from data collected from various environments, as shown in Fig. 1. Specifically, we first adopt Monte Carlo Tree Search (MCTS) to extract a common skeleton equation for

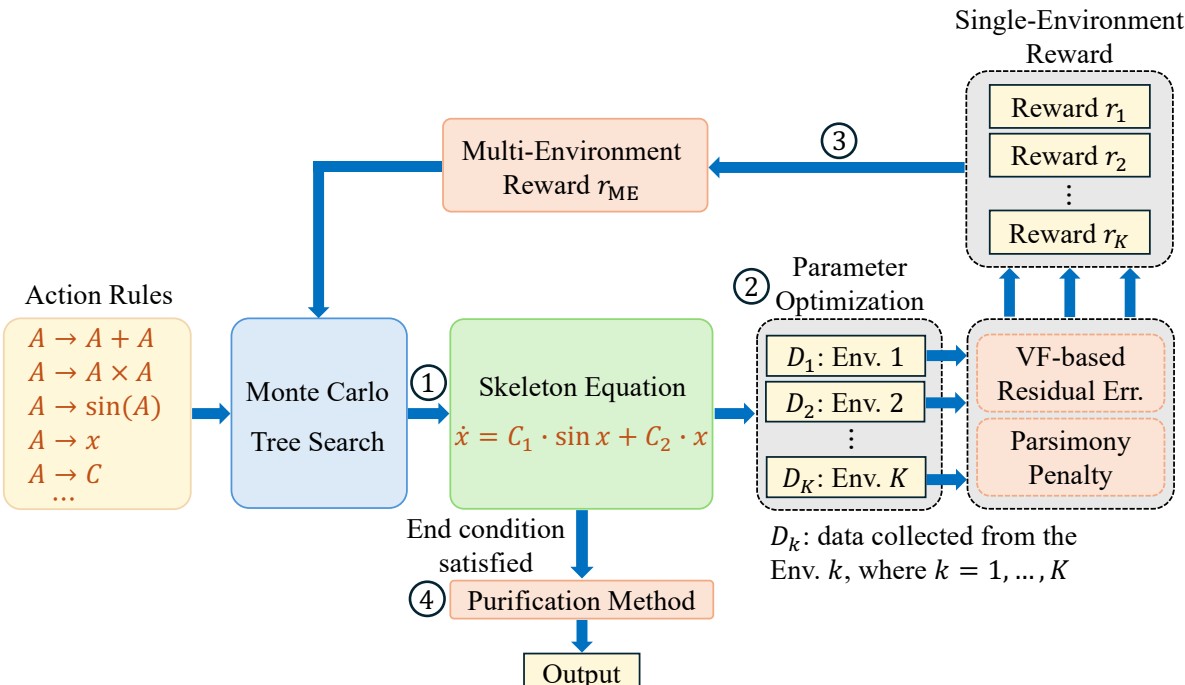

Figure 1: **The overall framework of the proposed IPAD.** It is performed in four main stages: **(1) Skeleton equation extraction.** It extracts a skeleton equation from the current expanded Monte Carlo Tree. **(2) Parameter optimization.** The skeleton is fed with data from various environments to optimize the parameters for each specific environment. **(3) Multi-Environment reward.** It derives a composite multi-environment rewards based on single-environment rewards from individual environments. **(4) Post-hoc purification method.** After the end condition is satisfied, the post-hoc purification method with prune the tiny coefficients of the discovered symbolic equations.

different environments at each step. Then we jointly optimize the system parameters for each specific environment based on their observed data. To enhance the estimation accuracy and robustness against noisy data, we adopt the Variational Formulation (VF) Courant et al. (1994), which implements *global integrals to avoid auto-regression*, to replace the L2-norm in the loss function. Meanwhile, we devise a novel multi-environment reward function for MCTS based on the designed VF loss. We further provide a theoretical analysis of the convergence rate of VF for symbolic regression. Besides, we devise a simple post-hoc purification method to remove tiny coefficients in the discovered equations for further improving discovery rate. Extensive experimental results on both simulated and real-world datasets demonstrate the superior performance of our method over baselines in identifying invariant physical laws across multiple environments.

In summary, our contributions include: (1) We propose the IPAD, a novel symbolic regression framework that identifies invariant physical dynamics from multiple environments. Our method can enhance the interpretability and generalization of ML for dynamical systems; (2) We devise a new VF-based multi-environment reward function to enhance robustness against noisy data and further theoretically analyze the VF convergence rate for symbolic regression; (3) A simple yet effective post-hoc purification method is developed to refine the discovered equations; (4) Extensive experiments demonstrate the superiority of IPAD over baseline methods, achieving a higher success rate in noisy and imbalanced data; (5) We collect a real-world damped pendulum dataset (see `https://anonymous.4open.science/r/Damped_Pendulum_Dataset-74D5`) from our physics laboratory to validate the effectiveness of the IPAD framework.

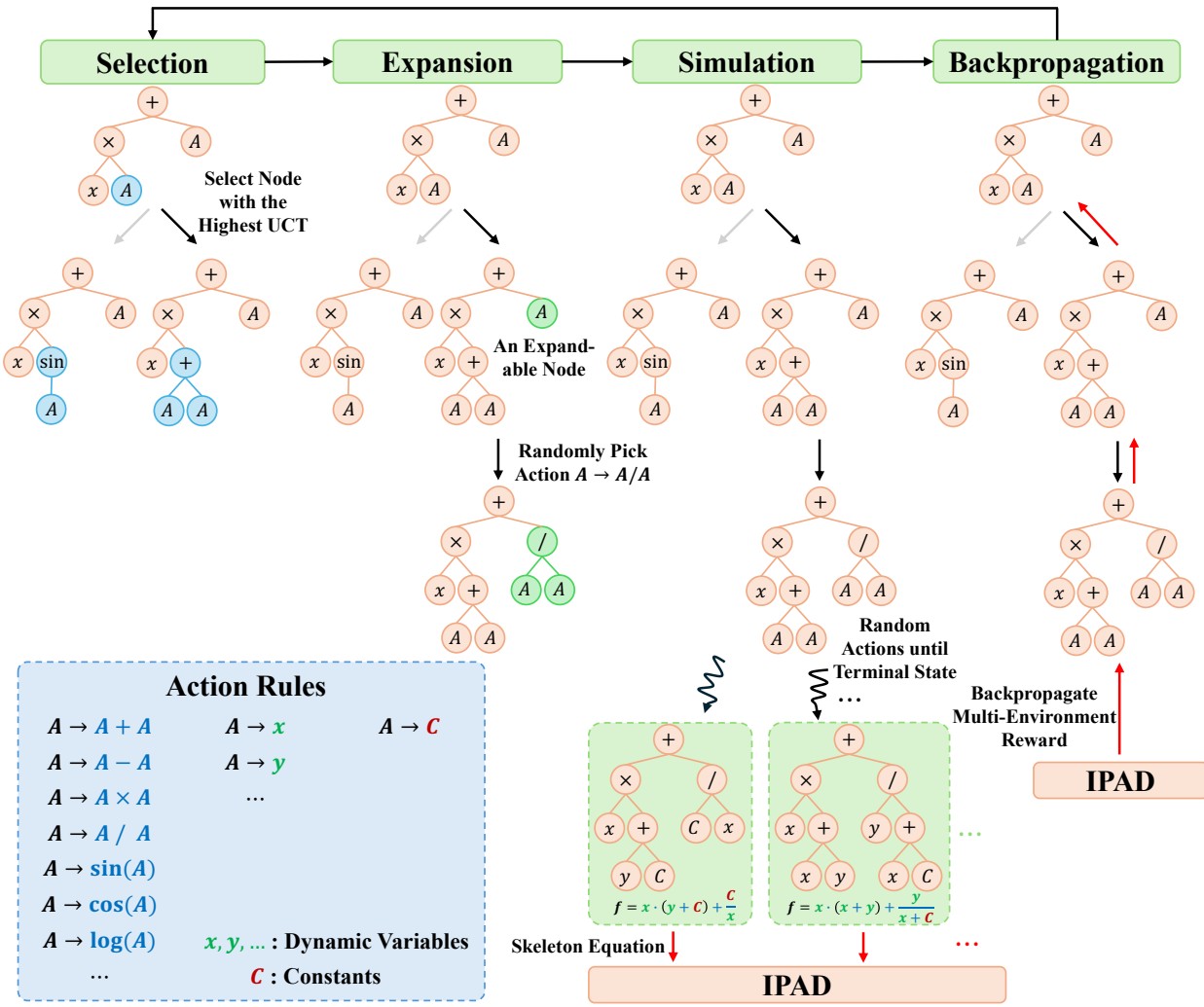

Figure 2: **Architecture of the MCTS for learning system dynamics.** An MCTS iteration consisting of four phases: selection, expansion, simulation, and backpropagation.

## 2 Related Work

**Single-Environment Symbolic Regression.** In earlier works, Genetic Programming (GP) was widely used in in symbolic regression Dubčáková (2011); Gustafson et al. (2005). Recent advancements integrated GP with reinforcement learning to enhance the performance Mundhenk et al. (2021); Petersen et al. (2019). Deep Neural Networks (DNNs) have also been employed, but initial models using trigonometric and exponential activations led to training instabilities Landajuela et al. (2021; 2022). AI-Feynman utilized a knowledge-based decomposition strategy heavily reliant on physical insights Udrescu & Tegmark (2020); Udrescu et al. (2020), while Deep Symbolic Regression (DSR) combined recurrent neural networks with reinforcement learning, offering improvements over GP methods yet struggling with complex equations Petersen et al. (2019). A recent work Sun et al. (2022) has employed Monte Carlo Tree Search (MCTS) to accelerate discovery using strong prior knowledge. Inspired by language modeling successes, pretraining techniques using Transformer architectures have been explored Valipour et al. (2021); Lample & Charton (2019); Biggio et al. (2021); Becker et al. (2023). However, all of these methods focus only on single-environment contexts, limiting their generalization capability across different environments. Moreover, these methods often struggle when data is sparse and noisy in real-world scenarios.

**Domain Adaptation across Multiple Environments.** Recent works have intensively studied domain adaptation methods to generalize ML models across multiple dynamical environments. Kirchmeyer et al. (2022) proposed an approach named context-informed dynamics adaptation (CoDA), which integrates specific environmental factors to enhance prediction accuracy across various settings. In addition, recent work Huang et al. (2023) introduced the Generalized Graph ODE (GG-ODE) framework, which combines Graph Neural Networks Scarselli et al. (2008) with ODEs to model continuous-time system dynamics across different environments. While these methods based on black-box DNNs can enhance model generalization, they often fail to explain and understand the underlying system behaviors. Most recent work has developed MetaPhysiCa Mouli et al. (2024) to improve the out-of-distribution robustness via a meta-learning procedure. However, it only discovers surrogate equations rather than the true invariant physical laws cross different environments.

## 3 Preliminaries

### 3.1 Problem Statement

The goal of this work is to identify invariant physical laws from data collected from multiple environments, allowing for fast adaptation to unseen environments. Suppose we collect data $\mathcal{D}$ with noise from $K$ environments, denoted by $\mathcal{D} = \{D_k\}_{k=1}^K$, where $D_k$ denotes the data samples collected from the $k$-th environment. For a given environment $k \in \{1, 2, \ldots, K\}$, its governing equations can be described by the following ODEs: $\dot{\boldsymbol{x}} = \boldsymbol{f}(\boldsymbol{x}(t); \boldsymbol{\theta}_k)$, where $\boldsymbol{x} = [x_1, \ldots, x_i, \ldots, x_N]^\top$ and $\boldsymbol{\theta}_k$ denotes the specific parameters for environment $k$. Based on the observed data $\mathcal{D}$ with some noise, we aim to identify a set of common skeleton equations $\boldsymbol{f} = [f_1, \ldots, f_i, \ldots, f_N]^\top$ with different constants across various environments. We define the important notations used in this work in Table 9.

### 3.2 Background of MCTS and VF

**Monte Carlo Tree Search (MCTS).** MCTS Chang et al. (2005); Kocsis & Szepesvári (2006); Coulom (2006) is a tree-based search algorithm used in various applications, including AlphaGo Fu (2016) and symbolic regression Sun et al. (2022). In MCTS, nodes represent partial solutions, edges represent actions or choices that extend the partial solution, and leaf nodes represent full solutions with associated rewards. Each iteration of MCTS consists of four phases: (1) **Selection**: Upper Confidence Bound for Trees (UCT) guides tree traversal from the root to either a leaf node or a node that has unvisited child nodes:

$$UCT(p, a) = Q(p, a) + c\sqrt{\frac{\ln M(p)}{M(p, a)}}, \tag{1}$$

where $Q(p, a)$ is the average reward for action $a$ at node $p$, $M(p)$ is $p$'s visit count, $M(p, a)$ is the selection count of $a$ at $p$, and $c$ controls exploration-exploitation balance; (2) **Expansion**: If the selected node is not a leaf and has unvisited child nodes, one such node is added to the tree; (3) **Simulation**: From the selected or newly added node, random actions are taken until a leaf is reached, repeated several times to estimate the node's value; (4) **Backpropagation**: The $Q$ and $M$ values are updated for all nodes from the evaluated node to the root, completing the iteration. This process is repeated for many iterations to build and refine the search tree.

**Variational Formulation (VF) for ODEs.** VF Hackbusch (2017) has the potential to eliminate intractable numerical differentiation errors caused by observation noise in evaluating ODEs.

**Theorem 1.** *(VF converges to L2, Tonti (1984); Qian et al. (2022)) Consider observed variable trajectories to be a continuously differentiable function $\boldsymbol{x}: [0, T] \to \mathbb{R}^N$, for some number of variables in the system $N \in \mathbb{N}^+$ and time horizon $T \in \mathbb{R}^+$. Let the continuous function $f_i: \mathbb{R}^N \to \mathbb{R}$ be the target function of the $i$-th environment variable $x_i$ such that $\dot{x}_i(t) = f_i(\boldsymbol{x}(t))$. Then for any Lipschitz continuous candidate function*

$\hat{f}_i$, let the VF loss be:

$$VF_S(\hat{f}_i, \boldsymbol{x}) = \sum_{s=1}^{S} \left( \int_0^T \hat{f}_i(\boldsymbol{x}(t))g_s(t) \ dt + \int_0^T x_i(t)\dot{g}_s(t) \ dt \right)^2, \tag{2}$$

where $\{g_1, g_2, \cdots\}$ is an orthonormal basis for the subspace of $L^2[0,T]$, and $S$ denotes the number of Hilbert orthogonal functions like sine functions.

Then

$$\lim_{S \to \infty} VF_S(\hat{f}_i, \boldsymbol{x}) = \|\hat{f}_i(\boldsymbol{x}(t)) - f_i(\boldsymbol{x}(t))\|_2^2 \tag{3}$$

Traditionally, L2 distance $\|\hat{f}_i(\boldsymbol{x}) - f_i(\boldsymbol{x})\|_2$ is used to evaluate how well a candidate function $\hat{f}_i$ fits a target ODE function $f_i$, for some trajectory $\boldsymbol{x}$. However, in real-world scenarios, $f_i(\boldsymbol{x})$ is not available, so it must be numerically derived from noisy observations $\boldsymbol{x}$, which incurs exceedingly high error relative to observation noise. In contrast, VF avoids this problem by approximating L2 distance without a need of $f_i(\boldsymbol{x})$.

### 3.3 MCTS for Learning System Dynamics

In this section, we detail the MCTS used to discover the underlying governing equations of dynamical systems, as shown in Fig. 2. The MCTS efficiently explores the space of possible mathematical expressions by constructing a search tree, where each node represents a partial or complete equation.

The input of the MCTS is a set of mathematical action rules, such as $A \to A + A$, $A \to \sin(A)$, $A \to C$, and $A \to x$, along with a list of dynamic variables $(x, y, \ldots)$. Here, $A$ is a nonterminal symbol representing an expression placeholder and $C$ denotes a constant. The interior nodes correspond to incomplete operations, while the leaf nodes represent variables (either dynamic variables or special variables). Actions, i.e., edges between nodes, are defined as the insertion of an operator or variable from the input set into the current incomplete equation.

In this work, MCTS involves the following four main steps.

(i) **Selection.** Starting from the root node (an empty equation), the tree is traversed following the highest Upper Confidence Bound for Trees (UCT) (Eq. 1) of available child nodes. The traversal continues until it reaches either a complete equation (a leaf node) or a node with unvisited child nodes (the parent of an unseen equation). The UCT score balances the exploitation of equations with good performance and the exploration of new equation structures.

(ii) **Expansion.** If the selected node represents an incomplete equation with unvisited child nodes, one of these unseen equations is randomly selected and added to the tree.

(iii) **Simulation.** Starting from the newly expanded node, operators and variables are randomly inserted to extend the equation until either a predefined depth is reached or a complete equation is generated. For example, an incomplete equation like $A + A$ might evolve into $A + A * A$, and eventually into $x + y * C$. The extracted equation is then fed into our IPAD framework, where parameters such as constants (e.g., $C$) are optimized using Powell's method, which will be detailed in the parameter optimization section. The fitness of the resulting equation is evaluated across multiple environments to compute a multi-environment reward.

(iv) **Backpropagation.** The statistics for all nodes along the path from the root to the evaluated equation are updated. For each node on this path, the visit count is increased, and the average reward is updated based on the obtained reward. This process refines the algorithm's understanding of promising equation structures and components, guiding future searches towards more optimal solutions.

## 4 Proposed IPAD

We present the overall framework of IPAD, as illustrated in Fig. 1. The IPAD framework is composed of four components: (1) skeleton equation extraction, (2) parameter optimization, (3) multi-environment reward,

and (4) post-hoc purification. In the following, we will detail each of these components, with particular focus on (3) and (4).

## 4.1 First Two Parts of IPAD

**Skeleton Equation Extraction (①).** We first use MCTS to extract the skeleton equation as shown in Fig. 1. The *skeleton equation* refers to the mathematical expression projected from the parse format of the tree with no parameters. More specifically, the input of the MCTS is a set of mathematical action rules, such as $A \to A + A$, $A \to \sin(A)$, $A \to C$, and $A \to x$, along with a list of state variables $(x, y, \ldots)$ of a dynamical system. Here, $A$ is a non-terminal symbol representing an expression placeholder and $C$ denotes a constant. During each MCTS iteration, we initiate by selecting and expanding a root node, and then extend the node until either a predefined depth is reached or a complete skeleton equation is generated. In this process, each MCTS action will add a system variable or operator to the skeleton equation. The detailed steps of obtaining the skeleton equation are presented in Appendix 3.3.

**Parameter Optimization (②).** Then the extracted skeleton equation is employed along with data from various environments to optimize the best parameters for each specific environment. We utilize Powell's method Mathews & Fink (2017) for the minimization search for the parameters. This step occurs during the simulation phase of MCTS, allowing us to evaluate the fitness (reward) of the equation for use in the backpropagation phase.

## 4.2 Multi-Environment Reward (③)

We first develop a single-environment reward strategy for dynamic systems characterized by distinct parameters. This strategy incorporates both residual error and a parsimony penalty to evaluate the symbolic expressions for each environment. Subsequently, an aggregator $\omega$ is employed to combine these single-environment rewards into a multi-environment reward, which is then utilized during the backpropagation phase of the MCTS. Further details on the reward functions are discussed below.

**Single-Environment Reward.** To assess the performance of each skeleton equation $f_i$, we opt for the Variational Formulation (VF) over the L2 norm to mitigate numerical errors. The *benefit of using VF* lies in implementing global integrals rather than relying on autoregressive steps in an ODE solver, which helps reduce error accumulation, especially in the presence of noisy data. The efficacy of VF, as compared to L2, will be substantiated in the ablation studies detailed in Section 5.3. VF is integrated into the MCTS to enhance the reward mechanism at each simulation step. As depicted in Fig. 1, the VF-based reward, incorporating a parsimony penalty for each candidate function $f_i$ is formulated for the $w$-th trajectory $\boldsymbol{x}_i^{(k,w)}$ of the $k$-th environment dataset $D_k$, given by:

$$r_k^{(w)} = \frac{\eta^n}{1 + \sqrt{\frac{1}{|D_k|} \min_{\boldsymbol{\theta}_k} \mathrm{VF}_S(f_i, \boldsymbol{x}_i^{(k,w)}; \boldsymbol{\theta}_k)}} \cdot \frac{1}{r_{\max}}, \tag{4}$$

where $|D_k|$ denotes the sample count, and the denominator minimizes the root-mean-square error (RMSE) utilizing VF (Eq. 2) instead of the squared L2 distance. The parsimony discount factor $\eta < 1$ and $n$, the number of nodes in the expression tree, are used in the numerator $\eta^n$ to penalize complex expressions. The factor $1/r_{\max}$ normalizes the $Q$-values for MCTS within the range $[0, 1]$, ensuring a consistent balance of exploration and exploitation where $r_{\max}$ is the highest reward observed during the search. When environment $k$ provides $W_k$ trajectories, the single-environment reward is the uniform mean of their per-trajectory rewards:

$$r_k = \frac{1}{W_k} \sum_{w=1}^{W_k} r_k^{(w)} \tag{5}$$

**Multi-Environment Reward.** Considering the potential imbalance across different environment datasets, we select the weighted mean as the aggregator $\omega$, factoring in the size of each dataset. Thus, the single-environment rewards are aggregated using $\omega$ into a comprehensive multi-environment reward, pivotal during

the backpropagation of MCTS. The formula for this multi-environment reward $r_{\mathrm{ME}}$ is given by:

$$r_{\mathrm{ME}} = \omega([r_k]_{k=1}^K) = \frac{\sum_{k=1}^K |D_k| \cdot r_k}{\sum_{k=1}^K |D_k|}, \tag{6}$$

where $r_k$ represents the single-environment reward from the $k$-th environment, and $|D_k|$ quantifies the dataset's size in each environment.

### 4.3 Post-hoc Purification Method (④)

Lastly, we identify the best-performing dynamic systems based on the highest multi-environment rewards. This purification step is triggered once the MCTS loop terminates - either when the multi-environment reward ceases to improve or when the number of iterations reaches the prescribed maximum. Due to the presence of noise in observed data, the discovered symbolic equations may include minor terms that effectively reduce residual errors but do not really represent the desired terms we seek. To address this issue, we propose a simple post-hoc purification technique aimed at removing these trivial terms. Such trivial terms are typically characterized by very small coefficients or negligible average values within the context of the entire equations.

Following the derivation of an analytical mathematical expression for an ODE variable from the MCTS process, denoted as $f(\boldsymbol{x}) = f(x_1, x_2, \ldots, x_N)$, where $N$ represents the dimension (number of variables in the ODEs), we proceed as follows:

First, on the $k$-th environment we extract the term set: $\mathcal{A}_{(k)} = \phi(f) = \{A_j | 1 \leq j \leq N_A\}$, where $N_A$ indicates the number of terms in the analytical mathematical expression $f$, and $A_j$ denotes the $j$-th term in the discovered equation. As mentioned above, the trivial terms often have negligible average values. Motivated by this, we propose to compute the average significance ratio $\overline{R}_j$ as follows:

$$\overline{R}_j = \frac{1}{|D_k|} \sum_{t \in [0,T]} \frac{|A_j(\boldsymbol{x}(t))|}{\sum_{l=1}^{N_A} |A_l(\boldsymbol{x}(t))|}, \tag{7}$$

where $j = 1, \ldots, N_A$ is the index of a term, $|D_k|$ represents the size of observed samples in the dataset of environment $k$, $T \in \mathbb{R}^+$ is the maximum time horizon we have data for, $A_j(\boldsymbol{x}(t))$ denotes the algebraic value obtained by substituting $\boldsymbol{x}(t)$ into the expression term $A_j$, and $\boldsymbol{x}(t)$ is a data vector from the dataset of a specific environment at time-step $t$.

Subsequently, we filter out the corresponding expression terms whose average significance ratio is smaller than a threshold $\tau$, resulting in a purified term set $\mathcal{A}'_{(k)}$:

$$\mathcal{A}'_{(k)} = \{A_j | \overline{R}_j \geq \tau, 1 \leq j \leq N_A\}. \tag{8}$$

We perform this procedure for each environment and identify the most frequent set of purified terms without coefficients, which we denote as $\mathcal{A}^*$. To adjust the parameters regarding the term adjustment, we repeat the parameter optimization process $\mathcal{P}$ introduced in Section 4.1 using the terms in $\mathcal{A}^*$ and the dataset of environment $k$ as:

$$\mathcal{A}^*_{(k)} = \mathcal{P}\left(\mathcal{A}^*, D_k\right). \tag{9}$$

Lastly, we obtain the symbolic expressions for each environment after purification. The detailed algorithm is presented in Algorithm 1 in Appendix B, and discussion on the selection of the hyperparameter $\tau$ are provided in Appendix I.

### 4.4 Theoretical Analysis of VF Convergence Rate

Within the IPAD framework, VF is critically employed to refine the optimization process, ensuring that the system dynamics are accurately estimated even under the challenging conditions of sparse and noisy datasets. Previous studies have established that VF converges to the L2 norm as the number of basis functions ($S$) approaches infinity. This provides a theoretical foundation but lacks practical insights on how quickly this convergence occurs. We further build upon previous results and prove VF convergence rate for sine basis in the following theorem.

**Theorem 2.** *(VF convergence rate) Consider observed trajectories to be a continuously differentiable function* $\boldsymbol{x}: [0,T] \to \mathbb{R}^N$, *for some number of variables in the system* $N \in \mathbb{N}^+$ *and time horizon* $T \in \mathbb{R}^+$. *Let the continuous function* $f_i: \mathbb{R}^N \to \mathbb{R}$ *be the true function of the* $i$-*th trajectory* $x_i$ *such that* $\dot{x}_i(t) = f_i(\boldsymbol{x}(t))$. *Suppose* $h(t) = \hat{f}_i(\boldsymbol{x}(t)) - f_i(\boldsymbol{x}(t)), h(t) \in C^\infty[0,T]$ *with bounded second derivatives* $\|h''(t)\|_2 < \infty$. *For any Lipschitz continuous function* $\hat{f}_i$ *that approximates* $f_i$, *let the VF loss be:*

$$
\begin{aligned}
VF_S(\hat{f}_i, \boldsymbol{x}(t)) &= \sum_{s=1}^{S} \left( \int_0^T \left( \hat{f}_i(\boldsymbol{x}(t)) - f_i(\boldsymbol{x}(t)) \right) g_s(t) dt \right)^2 \\
&= \sum_{s=1}^{S} \left( \int_0^T \hat{f}_i(\boldsymbol{x}(t)) g_s(t) \ dt + \int_0^T x_i(t) \dot{g}_s(t) \ dt \right)^2
\end{aligned}
\tag{10}
$$

*where* $\{g_1, g_2, \cdots\}$ *is the Fourier sine basis* $g_s = \sqrt{\frac{2}{T}} \sin\left(\frac{s\pi t}{T}\right)$ *of the space of square integrable functions with vanishing ends, i.e.* $f \in L^2[0,T]$ *where* $f(0) = f(T) = 0$. *We take the sum over* $S$ *functions from the above basis.*
*Then we have the following bound on the convergence rate of VF:*

$$
\|h(t)\|_2^2 - VF_S(\hat{f}_i, \boldsymbol{x}(t)) = O(1/S)
\tag{11}
$$

*In fact, this upper bound cannot be improved in general, due to the existence of function* $\hat{f}_i$ *such that*

$$
\exists \hat{f}_i, \ \|h(t)\|_2^2 - VF_S(\hat{f}_i, \boldsymbol{x}(t)) = \Theta(1/S)
$$

*We provide the detailed proof in Appendix K.*

**Remark 1.** *The theorem establishes a rigorous mathematical foundation for the use of VF in our IPAD framework. The demonstrated convergence rate of* $O(1/S)$, *while not rapid, underscores the practicality of employing VF, particularly in scenarios involving noisy and complex datasets. This motivates our choice of VF over traditional L2 norm methods. We also conduct experiments to verify this theorem in Section 5.3.*

**Remark 2.** *It is worth being explicit about what our reward actually guarantees. Recovering the true skeleton* $f^*$ *rests on three assumptions made by necessity: (i) given idealized MSE and optimal parameter, the penalized objective has* $f^*$ *as its unique maximizer (*identifiability*), which dictionary-free search cannot guarantee and which we therefore assume; (ii) MCTS locates this maximizer; and (iii) the per-environment parameters reach their global optimum under Powell's method. Of the three assumptions, (i) and (iii) are standard in free-form symbolic regression. The theorem 2 establishes the one component that closes the gap between idealized MSE* $(S \to \infty)$ *and the finite-S VF.*

# 5 Experiments

We first conduct extensive experiments to evaluate the effectiveness of the proposed IPAD using both simulated and real-world datasets under different environmental settings. Then, we implement ablation studies to investigate the important components in our method.

**Datasets.** (i) *Simulations:* We choose four dynamical systems: the Lotka-Volterra system Lotka (1925), the Damped Pendulum: system Strogatz (2018), the Lorenz systemLorenz (1962), and the SIR (Epidemiology) systemAnderson (1991). For an introduction to these dynamical systems, please refer to Appendix C.

To test the effectiveness and robustness of our model across different environmental dataset distributions, we choose $K = 5$ as the number of environments with different environment parameters and initial points, and select several distinct distributions of dataset trajectory number, including balanced and unbalanced distributions, as shown in Table 1. Please note that among Setting 2, 3 and 4, the total trajectory number of environments is fixed. We maintain this fixed trajectory capacity to observe how different dataset distributions affect the success rate.

For each environment, trajectories are generated using the LSODA method Hindmarsh (1983). Gaussian noise Papoulis (1965) is then applied to the observed data points to produce datasets with varying noise levels. We

Table 1: Dataset size distribution settings. Each environment shares the same system parameters but differs in initial conditions, leading to distinct trajectories. Settings 1, 2 are balanced distributions, while Settings 3,4 correspond to unbalanced ones. The total number of trajectories is fixed in Settings 2, 3, and 4.

| Distribution Setting | Balance Type | Number of Trajectories in Each Environment ($W_k$) | | | | | Total |
|---|---|---|---|---|---|---|---|
| | | Env. 1 | Env. 2 | Env. 3 | Env. 4 | Env. 5 | |
| Setting 1 | balanced | 10 | 10 | 10 | 10 | 10 | **50** |
| Setting 2 | balanced | 40 | 40 | 40 | 40 | 40 | **200** |
| Setting 3 | unbalanced | 120 | 20 | 20 | 20 | 20 | **200** |
| Setting 4 | unbalanced | 120 | 30 | 30 | 10 | 10 | **200** |

Table 2: Success rate of different methods on Lotka-Volterra System averaged over 20 random seeds

| Setting w/ | Setting 1 | | | | | Setting 2 | | | | | Setting 3 | | | | | Setting 4 | | | | |
|---|---|---|---|---|---|---|---|---|---|---|---|---|---|---|---|---|---|---|---|---|
| Noise Ratio | 0.00 | 0.05 | 0.10 | 0.15 | 0.20 | 0.00 | 0.05 | 0.10 | 0.15 | 0.20 | 0.00 | 0.05 | 0.10 | 0.15 | 0.20 | 0.00 | 0.05 | 0.10 | 0.15 | 0.20 |
| SPL(Avg) | 0.73 | 0.66 | 0.56 | 0.54 | 0.60 | 0.74 | 0.71 | 0.62 | 0.58 | 0.55 | 0.76 | 0.67 | 0.58 | 0.62 | 0.61 | 0.74 | 0.59 | 0.57 | 0.57 | 0.55 |
| SPL(W.V.) | 0.90 | 0.68 | 0.68 | 0.50 | 0.60 | 0.90 | 0.83 | 0.73 | 0.58 | 0.58 | 0.93 | 0.68 | 0.58 | 0.55 | 0.58 | 0.95 | 0.60 | 0.68 | 0.58 | 0.48 |
| D-code(Avg) | 0.07 | 0.04 | 0.01 | 0.03 | 0.01 | 0.14 | 0.09 | 0.07 | 0.04 | 0.04 | 0.09 | 0.06 | 0.04 | 0.03 | 0.03 | 0.12 | 0.07 | 0.04 | 0.04 | 0.02 |
| D-code(W.V.) | 0.08 | 0.08 | 0.05 | 0.05 | 0.05 | 0.15 | 0.13 | 0.10 | 0.05 | 0.05 | 0.15 | 0.08 | 0.05 | 0.05 | 0.05 | 0.15 | 0.08 | 0.05 | 0.05 | 0.05 |
| **IPAD** | **1.00** | **1.00** | **1.00** | **0.98** | **0.95** | **1.00** | **1.00** | **1.00** | **0.98** | **1.00** | **1.00** | **0.98** | **0.98** | **0.95** | **0.93** | **1.00** | **0.95** | **0.98** | **0.93** | **0.95** |

define a list of noise levels, $\Omega = \{0.00, 0.05, 0.10, 0.15, 0.20\}$, to assess how varying levels of noise influence the performance. For each noise ratio $\sigma_R \in \Sigma$, the noisy dataset is generated by $\boldsymbol{x}_{\text{noise}} = \boldsymbol{x}_{\text{origin}} + \boldsymbol{\epsilon} \cdot std(\boldsymbol{x}_{\text{origin}}) \cdot \sigma_R$, where $\boldsymbol{\epsilon} \sim \mathcal{N}(0,1)$ represents a random data array with a normal distribution, $std(\cdot)$ is the standard deviation method.

(ii) ***Real-world datasets:*** We also evaluate our method using two real-world applications: COVID19 Dataset and Damped Pendulum Dataset. Spefically, the COVID19 dataset was collected by JHU included in the Covsirphy Python library Takaya & Team (2020). We choose data samples from five countries starting in April 2020: *Armenia, France, Ireland, Spain, and the United Kingdom.* These data samples represent the dynamic data points of Susceptible (S), Infected (I) and Removed (R). Due to the limited quantity of observed daily data, we use 200 data points for each environment, collected over 200 consecutive days. Regarding Damped Pendulum Dataset, we collected the data samples using the equipment in a physics laboratory in our school. In our experiment, we have five distinct environments representing dynamics at different pendulum lengths. We will release our pendulum dataset upon publication. The detailed descriptions of these datasets are presented in C.5.

**Baselines.** We compare our method against the following baselines: (1) Symbolic Physics Learner (SPL) Sun et al. (2022), (2) D-code Qian et al. (2022), and (3) MetaPhysica Mouli et al. (2024). Note that MetaPhysica can only discover surrogate equations rather than true governing questions, leading to 0% success rate in our experiments. Thus, we report and explain its results in Appendix F.

Since some baseline methods are specifically designed for tasks in a single environment, where the unknown dynamic systems have fixed parameters, they do not natively support multi-environment tasks. Hence, we adopt two different strategies, average vote and weighted vote, to evaluate their performance under various environments. We detail these two strategies in Appendix D.

**Metrics.** Following prior works Qian et al. (2022); Sun et al. (2022), we use success rate, the average probability of successfully recovering the underlying dynamic systems' skeleton equation, as our evaluation metric.

## 5.1 Evaluation on Simulated Data

We first evaluate IPAD on four dynamical systems using simulated data. The parameters for dynamic systems and experimental settings are detailed in Appendix C and Appendix E, respectively.

Table 3: Success rate of different methods on the Lorenz System averaged over 20 random seeds.

| Setting w/ | Setting 1 | | | | | Setting 2 | | | | | Setting 3 | | | | | Setting 4 | | | | |
|---|---|---|---|---|---|---|---|---|---|---|---|---|---|---|---|---|---|---|---|---|
| Noise Ratio | 0.00 | 0.05 | 0.10 | 0.15 | 0.20 | 0.00 | 0.05 | 0.10 | 0.15 | 0.20 | 0.00 | 0.05 | 0.10 | 0.15 | 0.20 | 0.00 | 0.05 | 0.10 | 0.15 | 0.20 |
| SPL(Avg) | 0.56 | 0.59 | 0.50 | 0.44 | 0.37 | 0.53 | 0.56 | 0.51 | 0.50 | 0.45 | 0.52 | 0.55 | 0.49 | 0.48 | 0.42 | 0.53 | 0.57 | 0.49 | 0.49 | 0.39 |
| SPL(W.V.) | 0.53 | 0.62 | 0.50 | 0.42 | 0.37 | 0.48 | 0.55 | 0.47 | 0.48 | 0.48 | 0.55 | 0.55 | 0.57 | 0.48 | 0.47 | 0.55 | 0.58 | 0.50 | 0.57 | 0.37 |
| D-code(Avg) | 0.29 | 0.18 | 0.09 | 0.05 | 0.05 | 0.29 | 0.27 | 0.18 | 0.13 | 0.11 | 0.36 | 0.19 | 0.14 | 0.09 | 0.07 | 0.35 | 0.21 | 0.12 | 0.08 | 0.06 |
| D-code(W.V.) | 0.25 | 0.25 | 0.15 | 0.10 | 0.15 | 0.20 | 0.23 | 0.18 | 0.15 | 0.13 | 0.25 | 0.27 | 0.15 | 0.12 | 0.15 | 0.22 | 0.25 | 0.15 | 0.12 | 0.15 |
| **IPAD** | **0.80** | **0.73** | **0.73** | **0.80** | **0.73** | **0.75** | **0.73** | **0.73** | **0.70** | **0.73** | **0.77** | **0.75** | **0.77** | **0.80** | **0.73** | **0.72** | **0.72** | **0.75** | **0.70** | **0.72** |

Tables 2 and 3 compare the performance of various methods on the Lotka-Volterra and Lorenz systems. Results for the Damped Pendulum and SIR models are provided in Tables 14 and 15, respectively, in Appendix G. The results are averaged over 20 random seeds. Note that since the baselines (SPL and D-code) can only be applied to a single environment, there will be 200 ($20 \times 5$) results for them. We can observe from these tables that the proposed IPAD significantly outperforms the baselines under different dataset distribution settings in terms of success rate, especially under unbalanced distribution settings. Here the success rate of SPL drops significantly as the noise ratio increases, since it uses the squared L2 norm in the loss function. In contrast, our method is more robust to noisy data than SPL because of using VF loss. While D-code is robust to noisy data, its performance drops significantly when a single environment does not have enough data points. In summary, our IPAD exhibits superior performance over baselines in uncovering physical dynamics across multiple environments, particularly when data is noisy and imbalanced. Moreover, we conduct additional experiments to identify partially same but not exactly identical skeleton equations across different environments in Appendix H; we also describe how IPAD generalizes to a new, unseen environment when it is believed to share the same underlying dynamics as the training environments, as detailed in Appendix J.

## 5.2 Evaluation on Two Real-World Applications

In addition, we employ the proposed method to identify the underlying dynamics in two real-world applications: COVID19 Dataset and Damped Pendulum Dataset.

**(i) Evaluation on Real-World COVID19 Dataset**. We first assess the performance of our IPAD on COVID19 Dataset, as shown in Table 4. We can see that IPAD demonstrates superior performance over baseline methods in uncovering the underlying Equation 2 of the SIR model, which is the most complex and challenging equation of this model. The baselines do not work well since the real-world data collected is on a smaller scale than the datasets typically used in baseline studies. Moreover, the number of trajectories within the same environment is very limited (only one trajectory for each country), presenting significant challenges for the baselines in handling the noisy data. Note that, although our dataset contains only data samples without the ground-truth equations, the uncovered symbolic equation aligns very well with the SIR model, our assumed reference model.

**(ii) Real-World Damped Pendulum Dataset.** We also evaluate the performance of IPAD using the real-world damped pendulum dataset collected in the physics laboratory. The experimental results are reported in Table 5. Our method exhibits better performance in identifying the underlying Equation 2 of the damped pendulum system compared to the baselines. Based on the two real-world experiments, we conclude that IPAD demonstrates superiority in discovering invariant physical laws across multiple environments.

## 5.3 Ablation Studies

**Effect of number of basis functions used in the VF.** We also verify the VF convergence rate in Theorem 2. We use various candidate functions $\hat{f}_i$ to approximate the system's true dynamics, ranging from highly accurate fits to significantly deviating approximations. We do the experiments in a single environment setting with 500 samples and zero noise using the Lotka-Volterra system. We define the difference between VF and squared L2 residual error as $\Delta_S = \|h(t)\|_2^2 - \text{VF}_S(\hat{f}_i, \mathbf{x})$, where $S$ is the number of basis (sine) functions.

Table 4: **Experimental results on the real-world COVID19 dataset**. The results are reported using 20 different random seeds.

| Assumed Reference Eq. 2: $dI/dt$ | Method | Recovered Unified ODE Eq. | Appearance Rate | Success Rate |
|---|---|---|---|---|
| $C_1 \cdot SI + C_2 \cdot I$ | SPL | $C_1 \cdot I + C_2 \cdot R(+C_0)$ 
 $C_0$ 
 $C_1 \cdot SI + C_2 \cdot I(+C_0)$ 
 Others | 41.0% 
 18.0% 
 **3.0%** 
 25.0% | **3.0%** |
| | D-code | $C_0$ 
 $C_1 \cdot I(+C_0)$ 
 $C_1 \cdot S(+C_0)$ 
 Others | 75.0% 
 8.0% 
 4.0% 
 13.0% | **0.0%** |
| | **IPAD** | $C_1 \cdot SI + C_2 \cdot I(+C_0)$ 
 $C_1 \cdot IR + C_2 \cdot I(+C_0)$ 
 $C_1 \cdot SI + C_2 \cdot S + C_3 \cdot I(+C_0)$ 
 $C_1 \cdot I^2 + C_2 \cdot IR + C_3 \cdot I(+C_0)$ | **80.0%** 
 10.0% 
 5.0% 
 5.0% | **80.0%** |

Table 5: **Experimental results on the real-world damped pendulum dataset.** $\theta$ and $\omega$ denote the angle and angular velocity, respectively. Each environment consists of 500 data points. The results of IPAD are reported using 20 repeated experiments.

| Assumed Reference Eq. 2: $d\omega/dt$ | Method | Recovered Unified ODE Eq. | Appearance Rate | Success Rate |
|---|---|---|---|---|
| $C_1 \cdot \sin\theta + C_2 \cdot \omega$ | SPL | $C_1 \cdot \theta(+C_0)$ 
 $C_1 \cdot \sin\theta(+C_0)$ | 98.0% 
 2.0% | **0.0%** |
| | D-code | $C_1 \cdot \sin(\sin(\sin\theta))(+C_0)$ 
 $C_1 \cdot \sin\theta(+C_0)$ 
 $C_1 \cdot \sin\theta + C_2 \cdot \omega^2(+C_0)$ 
 Others | 13.0% 
 8.0% 
 8.0% 
 71.0% | **0.0%** |
| | **IPAD** | $C_1 \cdot \sin\theta + C_2 \cdot \omega(+C_0)$ 
 $C_1 \cdot \sin\theta(+C_0)$ 
 $C_1 \cdot \sin\theta + C_2 \cdot \sin\omega(+C_0)$ 
 $C_1 \cdot \sin\omega + C_2 \cdot \theta + C_3 \cdot \omega(+C_0)$ | **50.0%** 
 35.0% 
 10.0% 
 5.0% | **50.0%** |

Fig. 3 shows the theoretical prediction in Theorem 2 with the slope $\alpha \leq -1$, which is obtained through linear regression on the log-log data. However, divergence are noted in cases using the last two well-approximated equations. Divergence in the lower curves is primarily due to the small magnitude of $\Delta_S$. This becomes problematic as integration errors increase significantly. These errors escalate with the frequency of sine functions used in Simpson's rule Atkinson (1991). Besides, rounding errors compound the issue when contributions from high-frequency sine functions are minimal. When $\Delta_S$ falls below these growing numerical errors, VF will have an increasing difference with L2. In fact, this divergence impacts the success rate of IPAD, as illustrated in Table 6. As the number of basis functions is 50, our method has the highest success rate.

**Effectiveness of VF Loss.** We also study the impact of VF on the success rate of discovering physical dynamics under various noise levels. Table 7 illustrates the comparison results for the L2-norm vs. VF on the Damped Pendulum system using 20 random seeds. The outcomes reveal that substituting VF for the traditional L2-norm significantly enhances robustness, particularly in scenarios where the training dataset includes a certain degree of noise. This finding suggests that incorporating VF into our framework offers a potent solution to the challenges posed by noisy data.

**Effect of Purification Method.** We also explore the impact of the purification method on model performance. Table 8 presents the comparison results when the purification method is incorporated into the IPAD versus when it is not, using the SIR system. The results demonstrate that the purification method consistently enhances performance in both balanced and unbalanced distribution settings by effectively removing insignificant terms predicted by our method, thus improving the overall success rate.

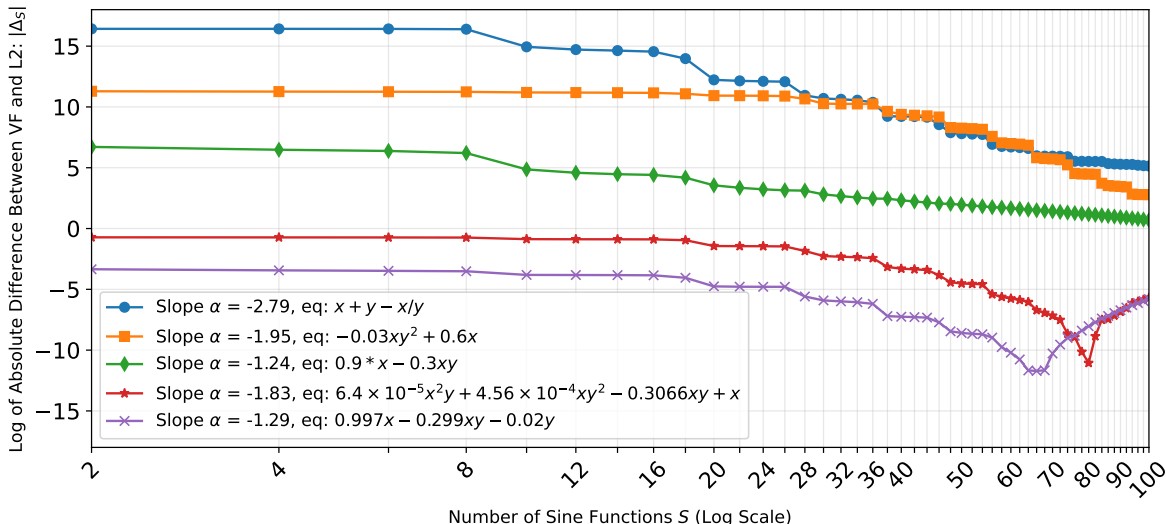

Figure 3: **Convergence of VF approximation error ($|\Delta_S|$) as a function of the number of sine functions ($S$) on a logarithmic scale.** A steeper negative slope indicates faster convergence of the approximation to the true residual error. All slopes $\alpha$ are less than $-1$, demonstrating alignment with the theoretical convergence rate of $O(S^{-1})$ (Theorem 2).

Table 6: Impact of number of basis functions on the success rate, with a fixed noise ratio of $\sigma_R = 0.2$ in the Damped Pendulum system.

| Settings | Purification | Number of Basis Functions $S$ | | | | | | | | | |
| | Method | 10 | 20 | 30 | 40 | 50 | 60 | 70 | 80 | 90 | 100 |
|---|---|---|---|---|---|---|---|---|---|---|---|
| Setting 2 (bal.) | Disabled | 0.78 | **0.88** | **0.88** | 0.83 | **0.88** | 0.80 | **0.88** | 0.78 | **0.88** | **0.88** |
| | Enabled | 0.90 | **0.95** | **0.95** | **0.95** | **0.95** | **0.95** | **0.95** | **0.95** | **0.95** | 0.93 |
| Setting 3 (unbal.) | Disabled | 0.83 | 0.83 | 0.88 | 0.85 | **0.90** | 0.73 | 0.85 | 0.88 | 0.83 | 0.88 |
| | Enabled | **0.95** | **0.95** | **0.95** | 0.93 | **0.95** | 0.88 | **0.95** | 0.93 | **0.95** | 0.93 |

Table 7: Impact of the VF loss on the success rate compared to L2-norm on the Damped Pendulum System.

| Dynamic System | Distribution Setting | Method | Noise Ratio $\sigma_R$ | | | | |
| | | | 0.00 | 0.05 | 0.10 | 0.15 | 0.20 |
|---|---|---|---|---|---|---|---|
| Damped Pendulum | Setting 1 (bal.) | L2 | **1.00** | **0.98** | 0.83 | 0.70 | 0.70 |
| | | VF | 0.98 | **0.98** | **0.95** | **0.93** | **0.90** |
| | Setting 2 (bal.) | L2 | **1.00** | 0.98 | 0.98 | 0.90 | 0.78 |
| | | VF | **1.00** | **1.00** | **1.00** | **0.98** | **0.93** |
| | Setting 3 (unbal.) | L2 | **1.00** | 0.95 | 0.95 | 0.83 | 0.73 |
| | | VF | **1.00** | **0.98** | **1.00** | **0.98** | **0.95** |
| | Setting 4 (unbal.) | L2 | **1.00** | 0.95 | 0.90 | 0.88 | 0.78 |
| | | VF | 0.98 | **1.00** | **0.98** | **0.98** | **0.80** |

## 6 Conclusion

This work proposed the IPAD, a novel symbolic regression framework to identify common physical laws across various environments. We integrated Variational Formulation (VF) into the multi-environment reward to enhance model performance. Furthermore, the devised purification method could successfully refine the discovered expressions via removing minor terms. Extensive experiments demonstrated our method can

Table 8: Impact of the purification method on success rate. Results are averaged over 20 random seeds for each experiment on the SIR system.

| Dynamic System | Distribution Setting | Purification Method | Noise Ratio $\sigma_R$ | | | | |
|---|---|---|---|---|---|---|---|
| | | | 0.00 | 0.05 | 0.10 | 0.15 | 0.20 |
| SIR | Setting 1 (balanced) | disabled | 0.733 | 0.883 | 0.833 | 0.767 | 0.733 |
| | | enabled | **1.000** | **1.000** | **1.000** | **0.967** | **0.950** |
| | Setting 2 (balanced) | disabled | 0.817 | 0.967 | 0.933 | 0.900 | 0.833 |
| | | enabled | **1.000** | **1.000** | **1.000** | **0.983** | **0.983** |
| | Setting 3 (unbalanced) | disabled | 0.833 | 0.967 | 0.883 | 0.850 | 0.800 |
| | | enabled | **1.000** | **1.000** | **1.000** | **0.983** | **0.967** |
| | Setting 4 (unbalanced) | disabled | 0.767 | 0.900 | 0.867 | 0.900 | 0.850 |
| | | enabled | **1.000** | **1.000** | **1.000** | **0.983** | **0.983** |

discover invariant physical dynamics with higher success rate than the baselines when data is noisy and sparse. We also discuss the limitations in Appendix L.

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

# A  Appendix: Summary of Notations

Table 9 describes the important notation used in our work.

Table 9: Summary of Notations.

| Notation | Definition |
|---|---|
| $K$ | The number of environments |
| $k$ | An environment index, $k \in \{1, 2, \ldots, K\}$ |
| $D_k$ | The data samples collected from the $k$-th environment |
| $|D_k|$ | The size of observed samples in the dataset of environment $k$ |
| $\mathcal{D}$ | All the collected data, $\mathcal{D} = \{D_k\}_{k=1}^K$ |
| $N$ | The number of variables in ODEs |
| $x_i$ | The $i$-th variable in ODEs, $i \in \{1, 2, \ldots, N\}$ |
| $\boldsymbol{x}$ | All the variables in ODEs, $\boldsymbol{x} = [x_1, \ldots, x_i, \ldots, x_N]^\top$ |
| $f_i$ | The target (ground-truth) ODE function corresponding to $i$-th variable, $i \in \{1, 2, \ldots, N\}$ |
| $\hat{f}_i$ | The candidate ODE function that approximates the target $f_i$ corresponding to $i$-th variable, $i \in \{1, 2, \ldots, N\}$ |
| $\boldsymbol{f}$ | The set of all equations in system: $\boldsymbol{f} = [f_1, \ldots, f_i, \ldots, f_N]^\top$ |
| $\boldsymbol{\theta}_k$ | The specific constants for all ODEs in environment $k$ |
| $UCT(p, a)$ | The upper confidence bound for trees (UCT) for choosing action $a$ at the current node $p$ |
| $Q(p, a)$ | The average reward for choosing action $a$ at the current node |
| $c$ | The exploration rate for controlling exploration and exploitation |
| $M(p)$ | The number of times node $p$ is visited |
| $M(p, a)$ | The number of times action $a$ is selected at node $p$ |
| $T$ | The number of samples in an environment dataset |
| $S$ | The number of basis functions in the VF method |
| $g_s$ | A basis function, index $s \in \{1, 2, \ldots, S\}$ |
| $r_k$ | The single-environment reward on the $k$-th environment $D_k$ |
| $r_{\mathrm{ME}}$ | The multi-environment reward |
| $r_{\max}$ | the highest reward observed during the search |
| $\eta$ | A parsimony discount factor, $\eta < 1$ |
| $n$ | The number of nodes in the expression tree, used in the parsimony penalty |
| $\omega(\cdot)$ | The weighted mean aggregator |
| $\mathcal{A}_{(k)}$ | The term set in the analytical mathematical format on the $k$-th environment |
| $\mathcal{A}^*$ | The most common purified term set among without coefficients |
| $\mathcal{A}_{(k)}^*$ | The recovered term set on the $k$-th environment |
| $N_A$ | The number of terms in an analytical mathematical term set $\mathcal{A}$ |
| $\phi(\cdot)$ | A method to extract all separate terms from a polynomial in the analytical form |
| $A_j$ | A term in an analytical mathematical term set $\mathcal{A}$, $1 \le j \le N_A$ |
| $\overline{R}_j$ | The average significance ratio of a term $A_j$ |
| $\tau$ | The threshold parameter used for filtering in the purification method |
| $\mathcal{P}(\cdot)$ | A parameter optimization process utilizing the Powell's method |
| $\Omega$ | The noise ratio set |
| $\sigma_R$ | A noise ratio, $\sigma_R \in \Sigma$ |
| $\Delta_S$ | The squared L2 residual error in VF, $\Delta_S = \|h(t)\|_2^2 - \mathrm{VF}_S(\hat{f}_i, \mathbf{x})$ |
| $\alpha$ | The steeper negative slope in VF convergence |
| $W_k$ | Number of trajectories |

# B   Appendix: Algorithm for the Purification Method

As introduced in Section 4.3, we propose a purification method to eliminate minor terms that effectively reduce residual errors, but do not really represent the desired terms we seek. The detailed algorithm is presented in Algorithm 1.

---

**Algorithm 1:** Purification Method

---

**Input:**

1. Observed samples $D_k$ in the dataset of environment $k$, each sample denoted as $\boldsymbol{x}(t)$;
2. Terms $\mathcal{A}_{(k)} = \{A_j | 1 \leq j \leq N_A\}$ from an equation discovered in the $k$-th environment;
3. Threshold parameter $\tau$, used for filtering;

**Output:** Purified term set $\mathcal{A}_{(k)}^*$;

**1** Initialize $\mathcal{A}_{(k)}^{'} \leftarrow \emptyset$;

**2 for** $j \leftarrow 1$ **to** $N_A$ **do**

**3**   Initialize $R_{\text{sum}} \leftarrow 0$;

**4**   **for** *observed sample $\boldsymbol{x}(t)$ from $D_k$* **do**

**5**     Substitute $\boldsymbol{x}(t)$ into the terms in $A$ and calculate $R_{\text{tmp}} \leftarrow \dfrac{|A_j(\boldsymbol{x}(t))|}{\sum_{l=1}^{N_A} |A_l(\boldsymbol{x}(t))|}$;

**6**     Update the sum: $R_{\text{sum}} \leftarrow R_{\text{sum}} + R_{\text{tmp}}$;

**7**   **end**

**8**   Compute the average significance ratio $\overline{R}_j \leftarrow \dfrac{R_{\text{sum}}}{|D_k|}$ for term $A_j$;

**9**   **if** $\overline{R}_j \geq \tau$ **then**

**10**    Insert $A_j$ into $\mathcal{A}_{(k)}^{'}$: $\mathcal{A}_{(k)}^{'} \leftarrow \mathcal{A}_{(k)}^{'} \cup \{A_j\}$;

**11**   **end**

**12 end**

**13** Perform this procedure for each environment and identify the most frequently occurring set of purified terms (without coefficients) among $\{\mathcal{A}_{(k)}^{'} | k = 1, \ldots, K\}$, which we denote as $\mathcal{A}^*$.

**14** Repeat the parameter optimization process $\mathcal{P}$: $\mathcal{A}_{(k)}^* \leftarrow \mathcal{P}(\mathcal{A}^*, D_k)$.

---

## C   Appendix: Dynamical Systems and Settings

### C.1   Lotka-Volterra (Predator–Prey) System

The Lotka-Volterra Lotka (1925) system, also known as the predator-prey system, describes the dynamics of two interacting species, predators and prey. The population change over time is governed by a pair of nonlinear differential equations:

$$\begin{cases} \dfrac{dx_1}{dt} = \alpha x_1 - \beta x_1 x_2 \\ \dfrac{dx_2}{dt} = -\gamma x_2 + \delta x_1 x_2 \end{cases} \tag{12}$$

In Eq. 12, $x_1$ and $x_2$ represent the population densities of prey and predators, respectively. The parameters $\alpha, \beta, \gamma, \delta$ are the rate constants for the birth of prey, killing because of predators, killing because of predators, and reproduction because of prey, respectively. An example simulation of the Lotka-Volterra system is shown in Fig. 4.

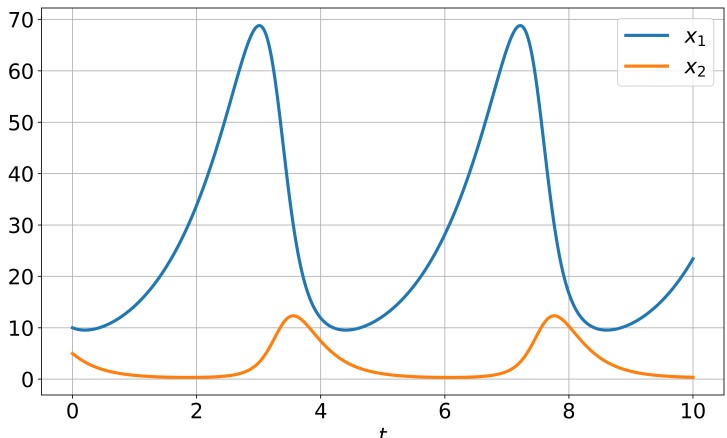

Figure 4: **Example dynamics of a Lotka-Volterra system.** This visualization showcases the dynamics of a Lotka-Volterra System. The parameters utilized in this example simulation are as follows: $\alpha = 1.00, \beta = 0.30, \gamma = 3.00, \delta = 0.10$. The initial population sizes for the prey ($x_1$) and predator ($x_2$) species are set to 10.0 and 5.0, respectively. The simulation unfolds over a time span of $[0, 10]$, illustrating the complex dynamics of predator-prey interactions over time.

In the Lotka-Volterra system, the set of environment parameters is denoted by $\boldsymbol{\theta} = \{\alpha, \beta, \gamma, \delta\}$. The sets of environment parameters used in our various environments are shown in Table 10.

Table 10: Lotka-Volterra System Settings

| Environment Id | Parameters | | | |
|---|---|---|---|---|
| | $\alpha$ | $\beta$ | $\gamma$ | $\delta$ |
| 1 | 1.00 | 0.30 | 3.00 | 0.10 |
| 2 | 1.20 | 0.39 | 2.80 | 0.09 |
| 3 | 1.30 | 0.42 | 3.20 | 0.08 |
| 4 | 1.10 | 0.51 | 3.10 | 0.11 |
| 5 | 0.90 | 0.39 | 2.90 | 0.12 |

### C.2 Damped Pendulum System

The Damped Pendulum system Strogatz (2018) is a classical mechanical model used to describe the motion of a pendulum under the influence of gravity and damping forces. It provides insights into oscillatory motion and energy dissipation caused by frictional effects in mechanical systems.

The dynamics of the damped pendulum system are governed by the following differential equations:

$$\begin{cases} \dfrac{dx_1}{dt} = x_2 \\ \dfrac{dx_2}{dt} = -\dfrac{g}{l}\sin(x_1) - k \cdot x_2 \end{cases} \tag{13}$$

In Eq. 13, $x_1$ represents the angular displacement (in radians) relative to the vertical equilibrium position, and $x_2$ represents the angular velocity. The parameter $g$ denotes the acceleration due to gravity, $l$ is the length of the pendulum, and $k$ is the damping coefficient that characterizes the frictional forces that act on the pendulum. The term $-\dfrac{g}{l}\sin(x_1)$ accounts for the restoring force due to gravity, while $-k \cdot x_2$ represents the damping force proportional to the angular velocity. An example simulation of the damped pendulum system is shown in Fig. 5.

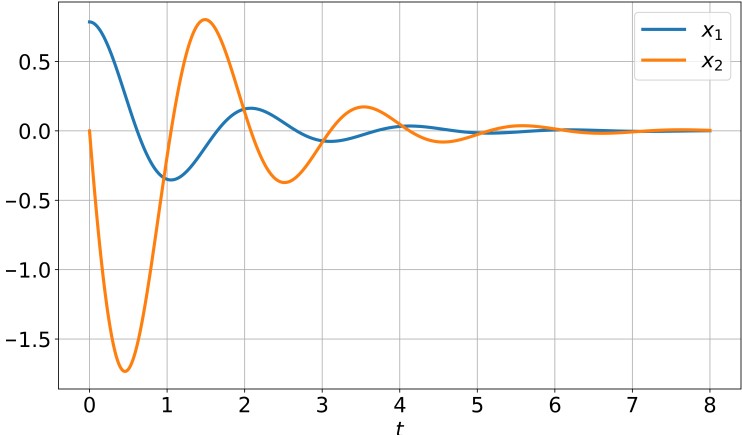

Figure 5: **Example dynamics of a Damped Pendulum system**. This visualization demonstrates the dynamics of a damped pendulum system. The parameters used in this example simulation are set as follows: $g = 10\text{m/s}^2$, representing the acceleration due to gravity; $l = 1.0\text{m}$, representing the length of the pendulum; $k = 1.5\text{s}^{-1}$, representing the damping coefficient. The initial values for the angular displacement $(x_1)$ and angular velocity $(x_2)$ are set to 0.5rad and 0rad/s, respectively. The simulation is conducted over a time domain of $[0, 8]$, illustrating the oscillatory motion of the pendulum under the influence of gravity and damping over time.

In the Damped Pendulum system, the set of environment parameters is denoted by $\boldsymbol{\theta} = \{l, k\}$. The environment parameter sets used in our various environments are shown in Table 11.

### C.3 Lorenz System

The Lorenz system Lorenz (1962) is a system of ordinary differential equations first studied by mathematician and meteorologist Edward Lorenz. It is now widely used in various scientific and engineering fields. The

Table 11: Damped Pendulum System Settings

| Environment Id | Parameters | |
|:--:|:--:|:--:|
| | $l$ | $k$ |
| 1 | 1.0 | 1.5 |
| 2 | 0.9 | 1.7 |
| 3 | 0.8 | 1.9 |
| 4 | 1.1 | 2.1 |
| 5 | 1.2 | 2.3 |

system consists of three ordinary differential equations:

$$
\begin{cases}
\dfrac{dx_1}{dt} = \sigma \left(x_2 - x_1\right) \\[2mm]
\dfrac{dx_2}{dt} = x_1 \left(\rho - x_3\right) - x_2 \\[2mm]
\dfrac{dx_3}{dt} = x_1 x_2 - \beta x_3
\end{cases}
\tag{14}
$$

In Eq. 14, $x_1$ is proportional to the rate of convection, $x_2$ to the horizontal temperature variation, and $x_3$ to the vertical temperature variation Sparrow (2012). The constants $\sigma$, $\rho$, and $\beta$ are system parameters proportional to the Prandtl number, Rayleigh number, and certain physical dimensions of the layer itself. An example simulation of the Lorenz system in shown in Fig. 6.

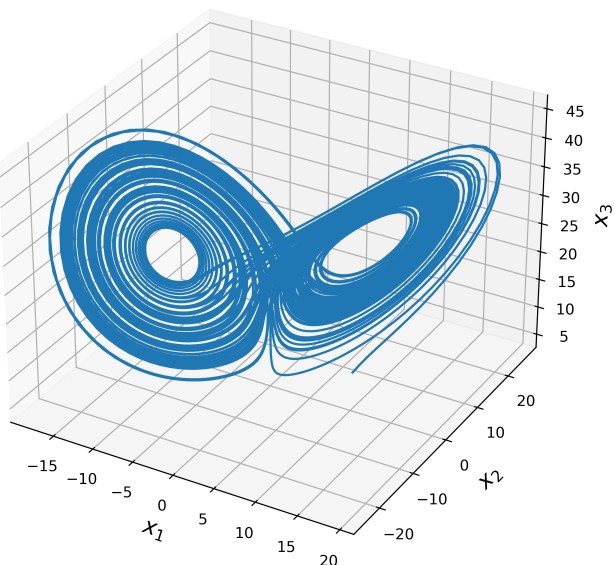

Figure 6: **Example dynamics of a Lorenz system.** This visualization illustrates the dynamics of the Lorenz system through a 3D plot. The parameters governing the example simulation are: $\sigma = 28.00, \rho = 10.00, \beta = 2.67$. Initial conditions for the $x_1$, $x_2$, and $x_3$ variables are set to 6.00, 6.00, and 15.00, respectively. This simulation unfolds over a time span of $[0, 10]$, showcasing the evolution of the Lorenz attractor in three-dimensional space.

In the Lorenz system, the set of environment parameters is denoted by $\boldsymbol{\theta} = \{\sigma, \rho, \beta\}$. The environment parameter sets used in our various environments are shown in Table 12.

Table 12: Lorenz System Settings

| Environment Id | Parameters | | |
| --- | --- | --- | --- |
| | $\rho$ | $\sigma$ | $\beta$ |
| 1 | 28.0 | 10.0 | 2.67 |
| 2 | 4.0 | 9.9 | 4.52 |
| 3 | 20.0 | 5.2 | 2.82 |
| 4 | 10.0 | 10.5 | 3.08 |
| 5 | 19.0 | 8.9 | 6.19 |

### C.4 SIR System

The SIR Anderson (1991) system is a classic compartmental system in epidemiology to simulate the spread of infectious disease. The basic SIR system considers a closed population with three different labels susceptible (S), infectious (I), and recovered (R). The evolution of the three interacting groups is predicted by the following equations:

$$\begin{cases} \dfrac{dx_1}{dt} = -\dfrac{\beta x_1 x_2}{N} \\ \dfrac{dx_2}{dt} = \dfrac{\beta x_1 x_2}{N} - \gamma x_2 \\ \dfrac{dx_3}{dt} = \gamma x_2 \end{cases} \tag{15}$$

In Eq. 15, $x_1$, $x_2$, and $x_3$ represent the susceptible, infected, and removed (either by death or recovery) populations, respectively. The constant value $N = x_1 + x_2 + x_3$ is the total population, $\beta$ refers to the contact rate between the susceptible and infected individuals, and $\gamma$ refers to the removal rate of the infected population. An example simulation of the SIR system in shown in Fig. 7.

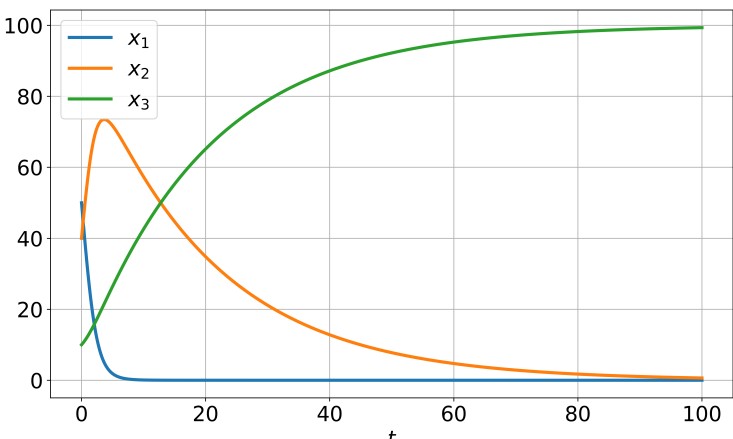

Figure 7: **Example dynamics of an SIR system.** This visualization demonstrates the dynamics of an SIR system. The parameters used in this example simulation are set as follows: $\beta = 0.01$, representing the transmission rate; $\gamma = 0.05$, representing the recovery rate. The initial values for the susceptible ($x_1$), infected ($x_2$), and removed ($x_3$) populations are set to 50.0, 40.0, and 10.0, respectively. The simulation is conducted over a time domain of $[0, 100]$, illustrating the progression of the disease within the population over time.

In the SIR system, the set of environment parameters is denoted by $\boldsymbol{\theta} = \{\beta, \gamma\}$. The environment parameter sets used in our various environments are shown in Table 13.

Table 13: SIR System Settings

| Environment Id | Parameters | |
| --- | --- | --- |
| | $\beta$ | $\gamma$ |
| 1 | 0.010 | 0.050 |
| 2 | 0.011 | 0.040 |
| 3 | 0.012 | 0.043 |
| 4 | 0.013 | 0.045 |
| 5 | 0.014 | 0.047 |

## C.5   Real-world SIR Dataset

The real-world SIR-COVID19 dataset Takaya & Team (2020) is provided via the Covsirphy Python library and includes daily counts of Confirmed (C), Infected (I), Fatal (F), and Recovered (Re) cases for 200 countries/regions. Since the system parameters may vary across countries and regions — for example, differences in the probability of disease transmission during contact between a susceptible and an infectious individual — we consider each country/region as a distinct environment. This implies that different environments are characterized by unique system parameters and initial conditions. In our experiment, we choose five countries as distinct environments: Armenia, France, Ireland, Spain, and the United Kingdom. Notably, the overall dataset exhibits data imbalance, with some countries or regions containing fewer than 50 data points. In this dataset, the number of Infected (I) is computed as Infected(I) = Confirmed (C) - Fatal (F) - Recovered (Re). Using the population (N) data for each country for the year 2020, we calculate the following:

$$
\begin{aligned}
\textbf{Susceptible(S)} &= \text{Population(N)} - \text{Confirmed(C)} \\
\textbf{Infected(I)} &= \text{Infected(I)} \\
\textbf{Removed(R)} &= \text{Fatal(F)} + \text{Recovered(Re)}
\end{aligned}
\tag{16}
$$

The calculated (S, I, R) dataset is used for our experiments, with a 1D Gaussian smoothing applied to reduce real-world noise. An example of the dynamics used in our experiments is shown in Fig. 8. The differential equations governing the SIR system are detailed in Appendix C.4.

We note that the COVID-19 results presented in this work are intended solely to demonstrate IPAD's ability to recover symbolic equation structure from real-world data, and should not be interpreted as a validated model for epidemiological forecasting or intervention design. Furthermore, the dataset is subject to inherent limitations including reporting bias, 1D Gaussian smoothing applied during preprocessing (see above), and time-varying disease dynamics that may not be fully captured by the fixed-parameter SIR formulation.

## C.6   Real-world Damped Pendulum Dataset

We perform experiments using five different pendulum lengths, with three repeated trials for each combination of settings, resulting in distinct dynamics. Data on both the angle and angular velocity of the pendulum are collected using the Wireless Rotary Motion Sensor (PS-3220), operated through PASCO software. The pendulum lengths are: (1) 0.236 m, (2) 0.330 m, (3) 0.426 m, (4) 0.518 m, and (5) 0.607 m. To simulate damping effects during motion, we mount a paper baffle on the pendulum. Our experiments use all five pendulum lengths as different environments, yielding a total of $5 \times 3 = 15$ unique dynamics. Fig. 9 (a) depicts the pendulum, mounted sensor, and laboratory environment.

The time axis is linearly scaled by a factor of 10 to moderate the dynamics gradients. To further refine the data, 1D Gaussian smoothing has been applied to reduce real-world noise. An example of the dynamics used in our experiments is illustrated in Fig. 9 (b), where $x_1$ (angular displacement (in radians) from the vertical equilibrium position) and $x_2$ (angular velocity) have been defined in Appendix C.2.

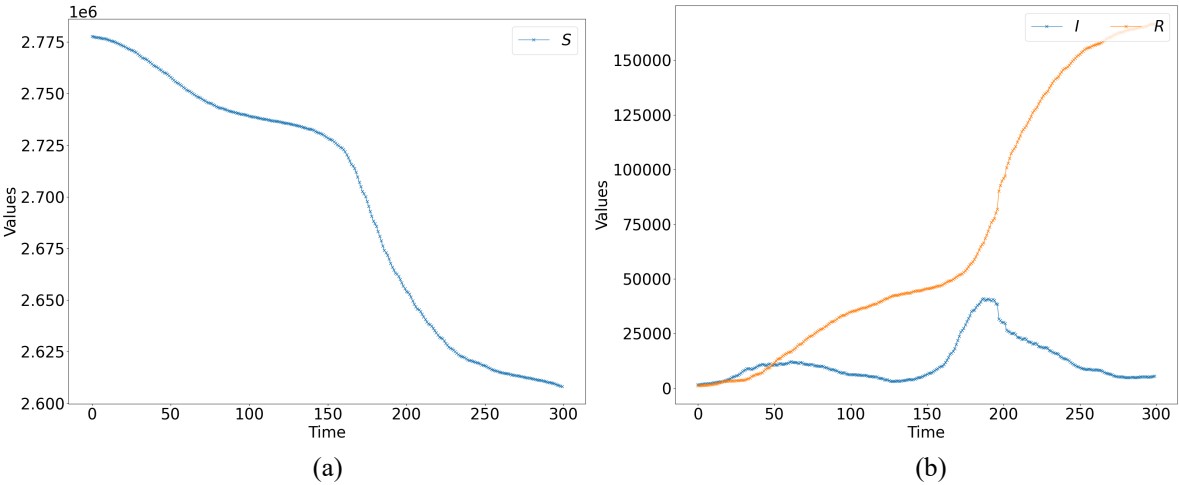

Figure 8: **Real-world SIR dataset**. Example dynamics from Armenia, spanning May 6, 2020, to March 1, 2021: (a) Dynamics of the S (Susceptible) population. (b) Dynamics of the I (Infected) and R (Removed) populations.

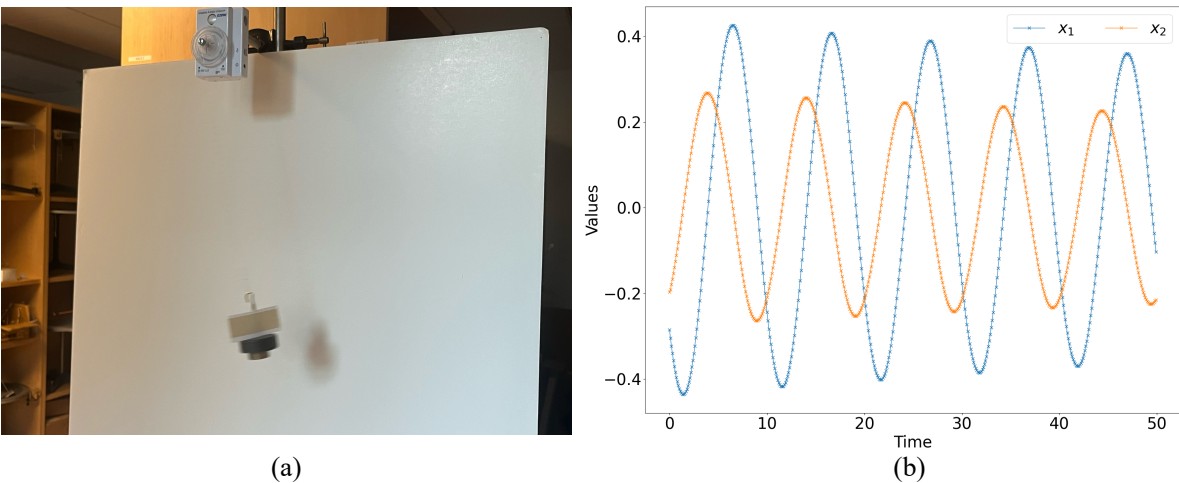

Figure 9: **Real-world Damped Pendulum dataset**. (a) The pendulum, mounted sensor, and laboratory environment. (b) Example dynamics of the damped pendulum system.

This dataset is publicly available through an anonymous repository at: `https://anonymous.4open.science/r/Damped_Pendulum_Dataset-60C4`.

# D   Appendix: Strategies for Applying Baselines to Multi-environment Tasks

Since the baseline methods, SPL and Dcode are tailored for tasks in a single environment with fixed parameters in the unknown dynamic systems, they inherently lack support for multi-environment tasks. To overcome this limitation, we have implemented two distinct strategies to assess performance across distributed datasets from various environments.

**Average (Avg).**   This approach is straightforward. We apply each baseline method separately to each environment $k$, resulting in $K$ distinct prediction outcomes. Each outcome is equally likely to be selected as the representative result for a specific dataset distribution setting. Thus, the performance for a given dataset distribution is the average of the results obtained in individual environments. In other word, in a test using a certain dataset distribution setting, the success rate under the average (Avg) strategy is as shown in Eq. 17:

$$B_{\text{avg}} = \frac{1}{K} \sum_{k=1}^{K} B_k \tag{17}$$

where $B_k \in \{0, 1\}$ is the prediction correctness on the single-environment dataset $D_k$.

**Weighted Vote (W.V.).**   The motivation of the weighed-vote strategy is that in settings where the sizes of individual environments vary, it is necessary to consider the size of each dataset. After collecting $K$ distinct prediction outcomes from each environment, we determine if at least half of the total dataset sizes support the correct results. This method ensures that the performance for a specific dataset distribution setting is determined through a weighted vote based on the single-environment results. In other word, in a test using a certain dataset distribution setting, the success rate under the weighted vote (W.V.) strategy is as shown in Eq. 18:

$$B_{\text{wv}} = \begin{cases} 1 & \text{if } \sum_{k=1}^{K} B_k \cdot |D_k| \geq \frac{1}{2} \sum_{k=1}^{K} |D_k|, \\ 0 & \text{otherwise.} \end{cases} \tag{18}$$

where the binary $B_k \in \{0, 1\}$ denotes the prediction correctness on the single-environment dataset $D_k$, and $|D_k|$ represents the size of observed samples in the dataset of environment $k$.

# E   Appendix: Experimental Settings

Following prior work Sun et al. (2022), both IPAD and SPL the MCTS exploration rate is set to $c = \frac{1}{\sqrt{2}} \approx 0.7071$. For each ODE, we run 500 episodes with a discount factor $\eta = 0.9999$ following prior work Sun et al. (2022) and a maximum state length of 50, beyond which the returned reward is 0. For the VF method in IPAD, we set the number of basis functions as $S = 50$.

For Dcode, we used the recommended sets of hyperparameters as specified in the original paper: population size: 15000, tournament size: 20, p-crossover: 0.6903, p-subtree mutation: 0.133, p-hoist mutation: 0.0361, p-point mutation: 0.0905, generations: 20, parsimony coefficient: 0.01. For a detailed explanation of genetic programming hyperparameters, please refer to Stephens (2021)

We run all experiments on an Ubuntu 20.04 machine equipped with an AMD EPYC 7543 32-Core Processor, 64 CPU threads (2 threads per core), 256 GB of memory, and four NVIDIA RTX A5000 GPUs, each with 24 GB of memory.

## F  Appendix: MetaPhysica Baseline: Experimental Results

We have compared our method against the latest baseline, MetaPhysica Mouli et al. (2024). We find that MetaPhysica consistently fails to accurately identify the invariant physical laws, achieving 0% success rate in all dynamical systems. This is primarily because MetaPhysica operates as a regression-based approximation of an equation, requiring predefined candidate terms (including all 0th, 1st, 2nd, and 3rd order algebraic terms, as well as basic sine and cosine terms by default). As a result, the approximated (surrogate) equations are unable to precisely capture the true underlying dynamics.

Below is an example of MetaPhysica's output for the case with a noise ratio of 0.00. The ground truth for equation 1 is:

$$\frac{dx_0}{dt} = C_1 \cdot x_0 + C_2 \cdot x_0 \cdot x_1$$

where $C_{1,2} = \{-0.39, 1.2\}$.

However, MetaPhysica uncovers the following equations:

$$\begin{aligned}
\frac{dx_0}{dt} = {} & C_1 \cdot x_0^3 + C_2 \cdot x_0^2 \cdot x_1 + C_3 \cdot x_0^2 + C_4 \cdot x_0 \cdot x_1^2 + C_5 \cdot x_0 \cdot x_1 \\
& + C_6 \cdot x_0 + C_7 \cdot x_1^3 + C_8 \cdot x_1^2 + C_9 \cdot x_1 + C_{10} \cdot \sin(x_0) \\
& + C_{11} \cdot \sin(x_1) + C_{12} \cdot \cos(x_0) + C_{13} \cdot \cos(x_1) + C_{14}
\end{aligned}$$

where the system coefficients are

$$\begin{aligned}
C_{1..14} = \{ & -0.000191, 0.001667, 0.019217, -0.013490, -0.157484, \\
& 0.616406, 0.016312, -0.224740, -0.210307, 0.026757, \\
& 0.705423, -0.175709, 0.745682, 1.461279\}.
\end{aligned}$$

In contrast, our IPAD achieves the same skeleton equation as the ground truth and outputs the predicted coefficients $C_{1,2} = \{-0.389936, 1.199808\}$.

# G  Appendix: Evaluation Results on Other Two Dynamical Systems

We also compare the performance of our method against the baselines on other two dynamical systems: Damped Pendulum and SIR. As shown in Tables 14 and 15, our IPAD significantly outperforms the baselines in identifying invariant physical laws from multiple environments.

Table 14: Comparison of methods on the Damped Pendulum System averaged over 20 random seeds.

| Setting w/ | Setting 1 | | | | | Setting 2 | | | | | Setting 3 | | | | | Setting 4 | | | | |
|---|---|---|---|---|---|---|---|---|---|---|---|---|---|---|---|---|---|---|---|---|
| Noise Ratio | 0.00 | 0.05 | 0.10 | 0.15 | 0.20 | 0.00 | 0.05 | 0.10 | 0.15 | 0.20 | 0.00 | 0.05 | 0.10 | 0.15 | 0.20 | 0.00 | 0.05 | 0.10 | 0.15 | 0.20 |
| SPL(Avg) | 0.83 | 0.64 | 0.42 | 0.32 | 0.26 | 0.79 | 0.80 | 0.72 | 0.61 | 0.60 | 0.77 | 0.73 | 0.61 | 0.47 | 0.51 | 0.79 | 0.70 | 0.62 | 0.45 | 0.51 |
| SPL(W.V.) | 0.83 | 0.75 | 0.28 | 0.20 | 0.13 | 0.83 | 0.85 | 0.83 | 0.60 | 0.65 | 0.88 | 0.75 | 0.83 | 0.75 | 0.63 | 0.70 | 0.78 | 0.68 | 0.68 | 0.63 |
| D-code(Avg) | 0.33 | 0.22 | 0.17 | 0.10 | 0.06 | 0.58 | 0.47 | 0.39 | 0.23 | 0.11 | 0.48 | 0.33 | 0.21 | 0.09 | 0.04 | 0.52 | 0.40 | 0.22 | 0.08 | 0.04 |
| D-code(W.V.) | 0.62 | 0.44 | 0.40 | 0.21 | 0.15 | 0.79 | 0.65 | 0.54 | 0.48 | 0.35 | 0.62 | 0.44 | 0.40 | 0.21 | 0.15 | 0.62 | 0.44 | 0.40 | 0.21 | 0.15 |
| **IPAD** | **0.98** | **0.98** | **0.95** | **0.93** | **0.90** | **1.00** | **1.00** | **1.00** | **0.98** | **0.95** | **1.00** | **0.98** | **1.00** | **0.98** | **0.95** | **0.98** | **1.00** | **0.98** | **0.98** | **0.80** |

Table 15: Comparison of methods on the SIR System averaged over 20 random seeds

| Setting w/ | Setting 1 | | | | | Setting 2 | | | | | Setting 3 | | | | | Setting 4 | | | | |
|---|---|---|---|---|---|---|---|---|---|---|---|---|---|---|---|---|---|---|---|---|
| Noise Ratio | 0.00 | 0.05 | 0.10 | 0.15 | 0.20 | 0.00 | 0.05 | 0.10 | 0.15 | 0.20 | 0.00 | 0.05 | 0.10 | 0.15 | 0.20 | 0.00 | 0.05 | 0.10 | 0.15 | 0.20 |
| SPL(Avg) | 0.80 | 0.62 | 0.53 | 0.42 | 0.31 | 0.79 | 0.81 | 0.68 | 0.66 | 0.57 | 0.79 | 0.73 | 0.65 | 0.55 | 0.49 | 0.81 | 0.73 | 0.62 | 0.53 | 0.45 |
| SPL(W.V.) | 0.82 | 0.67 | 0.52 | 0.37 | 0.20 | 0.82 | 0.87 | 0.75 | 0.63 | 0.57 | 0.73 | 0.82 | 0.72 | 0.62 | 0.52 | 0.82 | 0.85 | 0.83 | 0.67 | 0.58 |
| D-code(Avg) | 0.11 | 0.11 | 0.11 | 0.11 | 0.12 | 0.28 | 0.29 | 0.28 | 0.27 | 0.27 | 0.12 | 0.11 | 0.13 | 0.13 | 0.13 | 0.10 | 0.09 | 0.09 | 0.10 | 0.10 |
| D-code(W.V.) | 0.23 | 0.27 | 0.28 | 0.28 | 0.30 | 0.37 | 0.37 | 0.35 | 0.35 | 0.37 | 0.25 | 0.27 | 0.28 | 0.28 | 0.30 | 0.25 | 0.27 | 0.28 | 0.28 | 0.30 |
| **IPAD** | **1.00** | **1.00** | **1.00** | **0.97** | **0.95** | **1.00** | **1.00** | **1.00** | **0.98** | **0.97** | **1.00** | **1.00** | **1.00** | **0.98** | **0.97** | **1.00** | **1.00** | **1.00** | **0.98** | **0.98** |

# H    Appendix: More Experiments: Generalization to Partially Same Skeleton Equations Across Environments

In some scenarios, different environments may share partially same but not exactly identical skeleton equations. Our proposed IPAD framework still has the ability to predict such partially same or overlapping governing equations. This is because for terms that do not exist in some environments, IPAD tends to assign very small coefficients. The only additional step needed is to apply our purification method again, this time as a separate purification process tailored for each single environment, rather than using a unified purification method across environments. This adjustment is straightforward to implement.

To verify the feasibility of this approach, we selected the Friction Pendulum system (introduced in Appendix C.2) and its simpler form, the Non-Friction Pendulum system. These two systems share an identical Equation 1, differing only in Equation 2. Therefore, our experiments in this section focus solely on Equation 2. The two equations from Friction Pendulum and Non-Friction Pendulum system are shown in Eq. 19 and Eq. 20, respectively:

$$\frac{dx_2}{dt} = -\frac{g}{l}\sin(x_1) - k \cdot x_2 \tag{19}$$

$$\frac{dx_2}{dt} = -\frac{g}{l}\sin(x_1), \tag{20}$$

where $x_1$ represents the angular displacement (in radians) relative to the vertical equilibrium position, and $x_2$ represents the angular velocity. The parameter $g$ denotes the acceleration due to gravity, $l$ is the length of the pendulum, and $k$ is the damping coefficient that characterizes the frictional forces that act on the pendulum.

We first set up a simple experimental configuration: with number of environment $K = 5$, noise ratio $\delta_R = 0.00$, and under Setting 2 (balanced distribution, see Section 5), we configure two of the five environments as Non-Friction Pendulum systems (without friction term), and the remaining three as Friction Pendulum systems (with friction term). The ground truth equations are:

Non-Friction Pendulum system:

$$\begin{cases}
\text{Env 1: } dx_2/dt = -10.0000 \cdot \sin(x_1) - 1.5000 \cdot x_2 \\
\text{Env 2: } dx_2/dt = -11.1111 \cdot \sin(x_1) \\
\text{Env 3: } dx_2/dt = -12.5000 \cdot \sin(x_1) \\
\text{Env 4: } dx_2/dt = -9.0909 \cdot \sin(x_1) - 2.1000 \cdot x_2 \\
\text{Env 5: } dx_2/dt = -8.3333 \cdot \sin(x_1) - 2.3000 \cdot x_2
\end{cases}$$

After running IPAD once, we obtained the following results:

$$\begin{cases}
\text{Env 1: } dx_2/dt = -9.9972 \cdot \sin(x_1) - 1.4994 \cdot x_2 \\
\text{Env 2: } dx_2/dt = -11.1093 \cdot \sin(x_1) + 1.8208e^{-6} \cdot x_2 \\
\text{Env 3: } dx_2/dt = -12.4976 \cdot \sin(x_1) + 2.9643e^{-4} \cdot x_2 \\
\text{Env 4: } dx_2/dt = -9.0874 \cdot \sin(x_1) - 2.0990 \cdot x_2 \\
\text{Env 5: } dx_2/dt = -8.3287 \cdot \sin(x_1) - 2.2984 \cdot x_2
\end{cases}$$

We then apply the single-environment purification method, which removes terms with minor average weights from the equation and subsequently re-optimizes the parameters. As a result, we obtain the following:

$$\begin{cases} \text{Env 1: } dx_2/dt = -9.9972 \cdot \sin(x_1) - 1.4994 \cdot x_2 \\ \text{Env 2: } dx_2/dt = -11.1151 \cdot \sin(x_1) \\ \text{Env 3: } dx_2/dt = -12.4981 \cdot \sin(x_1) \\ \text{Env 4: } dx_2/dt = -9.0874 \cdot \sin(x_1) - 2.0990 \cdot x_2 \\ \text{Env 5: } dx_2/dt = -8.3287 \cdot \sin(x_1) - 2.2984 \cdot x_2 \end{cases}$$

*To further evaluate IPAD's performance under more complex and challenging settings*, we vary the noise ratio $\delta_R$ (from 0.00 to 0.20) and test both Setting 2 and Setting 3 (balanced and unbalanced distribution, respectively). For each condition, we perform 40 experiments using different random seeds. The results are summarized in Table 16.

Table 16: Success rate of predicting correct equation skeleton under different noise ratios and settings over 40 random seeds

| Settings | Noise Ratio $\delta_R$ | | | | |
| --- | --- | --- | --- | --- | --- |
| | 0.00 | 0.05 | 0.10 | 0.15 | 0.20 |
| Setting 2 (bal.) | 1.00 | 0.95 | 0.88 | 0.75 | 0.73 |
| Setting 3 (unbal.) | 0.98 | 1.00 | 0.85 | 0.78 | 0.68 |

From these experimental results, we confirm that IPAD exhibits a certain inherent ability to predict similar skeleton equations across different environments. This ability is validated on both balanced and unbalanced dataset distributions, and especially effective under lower noise ratios.

# I Appendix: Important Hyper-parameters Tuning

As discussed in Section 4.3, the purification method requires setting a threshold hyperparameter, $\tau$, to filter out insignificant terms, with the optimal value potentially varying across different systems. We assess the average significance ratio of matched versus unmatched terms in each identified equation to determine the most suitable $\tau$. Our experiments are conducted on the SIR system, with results averaged over three dynamic variables (S, I, R) and 20 random seeds.

Our experiments are conducted on the SIR system, with results averaged over three dynamic variables (S, I, R) and 20 random seeds. This hyperparameter selection was performed independently of, and prior to, the main evaluation experiments reported in Tables 2–3 (Section 5) and Tables 14–15 (Appendix G); the resulting fixed value of $\tau$ (and similarly $S$, tuned in Section 5.3) was then applied uniformly across all reported results, with no further tuning on test outcomes.

Table 17 shows that a wide range of $\tau$ performs well for the SIR system, particularly at low noise ratios (e.g., 0.00 or 0.05), where thresholds from $\tau = 0.16$ to $\tau = 0.30$ consistently achieve a success rate of 1.000. This indicates that the purification method is robust to the choice of $\tau$ when the noise level is low.

A key observation is that setting $\tau$ too low may fail to remove noise terms with small average weights, while choosing a higher $\tau$ could mistakenly eliminate important terms in the equations. Considering these factors, we select $\tau = 0.20$ as a balanced choice for the SIR system, ensuring effective noise filtering without discarding significant terms. This value provides a good compromise between maintaining a high success rate and avoiding excessive filtering across different noise settings.

Table 17: Impact of the filter parameter on success rate. Results are averaged over 20 random seeds for each experiment on the SIR system.

| Distribution Setting | Noise Ratio | Filter Parameter $\tau$ | | | | | | | | |
|---|---|---|---|---|---|---|---|---|---|---|
| | | 0.16 | 0.18 | 0.20 | 0.22 | ... | 0.28 | 0.30 | 0.32 | 0.34 |
| Setting 1 | 0.00 | **1.000** | **1.000** | **1.000** | **1.000** | ... | **1.000** | **1.000** | **1.000** | 0.983 |
| | 0.05 | **1.000** | **1.000** | **1.000** | **1.000** | ... | **1.000** | **1.000** | **1.000** | 0.983 |
| | 0.10 | 0.983 | **1.000** | **1.000** | **1.000** | ... | **1.000** | **1.000** | **1.000** | 0.983 |
| | 0.15 | 0.933 | 0.950 | **0.967** | **0.967** | ... | **0.967** | **0.967** | **0.967** | 0.933 |
| | 0.20 | 0.900 | 0.917 | **0.950** | **0.950** | ... | **0.950** | **0.950** | **0.950** | 0.917 |
| Setting 2 | 0.00 | **1.000** | **1.000** | **1.000** | **1.000** | ... | **1.000** | **1.000** | **1.000** | **1.000** |
| | 0.05 | **1.000** | **1.000** | **1.000** | **1.000** | ... | **1.000** | **1.000** | **1.000** | **1.000** |
| | 0.10 | **1.000** | **1.000** | **1.000** | **1.000** | ... | **1.000** | **1.000** | **1.000** | **1.000** |
| | 0.15 | **0.983** | **0.983** | **0.983** | **0.983** | ... | **0.983** | **0.983** | **0.983** | **0.983** |
| | 0.20 | **0.983** | **0.983** | **0.983** | **0.983** | ... | **0.983** | **0.983** | **0.983** | **0.983** |
| Setting 3 | 0.00 | **1.000** | **1.000** | **1.000** | **1.000** | ... | **1.000** | **1.000** | 0.983 | 0.967 |
| | 0.05 | **1.000** | **1.000** | **1.000** | **1.000** | ... | **1.000** | **1.000** | 0.983 | 0.983 |
| | 0.10 | **1.000** | **1.000** | **1.000** | **1.000** | ... | **1.000** | **1.000** | **1.000** | 0.983 |
| | 0.15 | **0.983** | **0.983** | **0.983** | **0.983** | ... | **0.983** | **0.983** | **0.983** | 0.967 |
| | 0.20 | **0.967** | **0.967** | **0.967** | **0.967** | ... | **0.967** | **0.967** | **0.967** | **0.967** |
| Setting 4 | 0.00 | **1.000** | **1.000** | **1.000** | **1.000** | ... | **1.000** | **1.000** | 0.983 | 0.983 |
| | 0.05 | **1.000** | **1.000** | **1.000** | **1.000** | ... | **1.000** | **1.000** | 0.983 | 0.983 |
| | 0.10 | **1.000** | **1.000** | **1.000** | **1.000** | ... | **1.000** | **1.000** | 0.983 | 0.983 |
| | 0.15 | **0.983** | **0.983** | **0.983** | **0.983** | ... | **0.983** | **0.983** | 0.967 | 0.967 |
| | 0.20 | 0.950 | **0.983** | **0.983** | **0.983** | ... | **0.983** | **0.983** | 0.967 | 0.967 |

## J  Appendix: Generalization to Unseen Environments

IPAD is designed under the assumption that data collected from multiple environments share a common underlying differential equation skeleton, while environment-specific dynamics are captured through different parameter values. This assumption naturally enables IPAD to generalize to a new, unseen environment, provided that the new environment is believed to share the same governing skeleton as the environments used during structure discovery.

**Setup.**   Suppose IPAD has already discovered a shared equation skeleton $\hat{f}$ from a set of training environments $\mathcal{E}_{\text{train}} = \{E_1, \ldots, E_K\}$ (Step 1 of Figure 1), together with environment-specific parameters $\{\theta_1, \ldots, \theta_K\}$ obtained via parameter optimization (Step 2 of Figure 1). Given a new environment $E_{K+1}$ with its own observed trajectory data, and assuming $E_{K+1}$ shares the same skeleton $\hat{f}$, generalization to $E_{K+1}$ requires only estimating its environment-specific parameters $\theta_{K+1}$.

**Procedure.**   This reduces to a single additional parameter optimization pass, identical in form to the procedure already applied independently to each of the $K$ training environments in Step 2. Concretely:

1. Fix the skeleton $\hat{f}$ discovered from $\mathcal{E}_{\text{train}}$.

2. Using the observed data from $E_{K+1}$, optimize $\theta_{K+1}$ to minimize the same objective used in Step 2 (e.g., trajectory-fitting loss) for the fixed skeleton $\hat{f}$.

3. No re-execution of the structure discovery phase (Step 1) is required.

This procedure is computationally lightweight relative to full structure discovery, since parameter optimization over a fixed skeleton is a well-established and efficient technique, and it has already been performed $K$ times (once per training environment) as part of the main experiments reported in Section 5.

**Discussion.**   The key insight is that the transferable component of IPAD is the equation skeleton itself, not the environment-specific parameters. Because the skeleton is environment-agnostic by construction, generalization to a new environment under the shared-skeleton assumption does not require new architectural changes or repeated structure search, which is a direct consequence of how IPAD decomposes the discovery problem into structure recovery (Step 1) and parameter fitting (Step 2).

We note that this generalization procedure presumes the shared-skeleton assumption holds for the new environment. If the unseen environment instead follows a partially different skeleton (e.g., containing additional or missing terms relative to the training environments), the relevant procedure is the one described in Appendix H, where structure discovery is still required, but our purification method can help recover the partially overlapping skeleton across environments.

# K   Appendix: Proof of Theorem 2: VF Convergence Rate

**Theorem 3.** *(VF convergence rate) Consider observed trajectories to be a continuously differentiable function $\boldsymbol{x}: [0, T] \to \mathbb{R}^N$, for some number of variables in the system $N \in \mathbb{N}^+$ and time horizon $T \in \mathbb{R}^+$. Let the continuous function $f_i: \mathbb{R}^N \to \mathbb{R}$ be the true function of the i-th trajectory $x_i$ such that $\dot{x}_i(t) = f_i(\boldsymbol{x}(t))$. Suppose $h(t) = \hat{f}_i(\boldsymbol{x}(t)) - f_i(\boldsymbol{x}(t)), h(t) \in C^\infty[0, T]$ with bounded second derivatives $\|h''(t)\|_2 < \infty$. For any Lipschitz continuous function Folland (1999) $\hat{f}_i$ that approximates $f_i$, let the VF loss be:*

$$VF_S(\hat{f}_i, \boldsymbol{x}(t)) = \sum_{s=1}^{S} \left( \int_0^T \left( \hat{f}_i(\boldsymbol{x}(t)) - f_i(\boldsymbol{x}(t)) \right) g_s(t) dt \right)^2$$

$$= \sum_{s=1}^{S} \left( \int_0^T \hat{f}_i(\boldsymbol{x}(t)) g_s(t) \ dt + \int_0^T x_i(t) \dot{g}_s(t) \ dt \right)^2 \tag{21}$$

*where $\{g_1, g_2, \cdots\}$ is the Fourier sine basis Churchill (1941) $g_s = \sqrt{\frac{2}{T}} \sin\left(\frac{s\pi t}{T}\right)$ of the space of square integrable functions with vanishing ends, i.e. $f \in L^2[0, T]$ where $f(0) = f(T) = 0$. We take the sum over $S$ functions from the above basis. Then we have the following bound on the convergence rate of VF:*

$$\|h(t)\|_2^2 - VF_S(\hat{f}_i, \boldsymbol{x}(t)) = O(1/S) \tag{22}$$

*In fact, this upper bound cannot be improved in general, due to the existence of function $\hat{f}_i$ such that*

$$\exists \hat{f}_i, \ \|h(t)\|_2^2 - VF_S(\hat{f}_i, \boldsymbol{x}(t)) = \Theta(1/S) \tag{23}$$

*Proof.* Consider the inner product of the Hilbert space $L^2[0, T]$ and its norm

$$\langle h, g_s \rangle = \int_0^T h(t) g_s(t) dt \quad \text{and} \quad \|h\|_2^2 = \langle h, h \rangle \tag{24}$$

where $g_s(t) = \sqrt{\frac{2}{T}} \sin\left(\frac{s\pi t}{T}\right)$ for $s = 1, 2, \ldots$ form an orthonormal basis for $L^2[0, T]$.

For $h \in L^2[0, T]$, we can consider its Fourier sine approximation $\sum_{s=1}^{\infty} \langle h, g_s \rangle g_s(t)$. Note that the Fourier sine series work on discontinuous functions Ulanowicz (1997). To understand the relationship between $h(t)$ and this approximation, let's consider a function $k(t)$ defined as:

$$k(t) = \begin{cases} h(t) & \text{if } 0 < t < T \\ 0 & \text{if } t = 0 \text{ or } t = T \end{cases} \tag{25}$$

Note that $k(t)$ differs from $h(t)$ only at the endpoints. Since $h(t) g_s(t)$ and $k(t) g_s(t)$ differ only at a finite number of points $0, T$, and the integral is not affected by changes on a set of measure zero (Theorem 7.1.7 in Aguilar (2022)), we have

$$\langle k, g_s \rangle = \int_0^T k(t) g_s(t) dt = \int_0^T h(t) g_s(t) dt = \langle h, g_s \rangle \tag{26}$$

In other words, their Fourier sine coefficients are identical. Therefore,

$$h(t) = k(t) = \sum_{s=1}^{\infty} \langle k, g_s \rangle g_s(t) = \sum_{s=1}^{\infty} \langle h, g_s \rangle g_s(t) \tag{27}$$

at almost every $t$. This fact implies

$$\|h(t)\|_2^2 = \left\| \sum_{s=1}^{\infty} \langle h, g_s \rangle g_s(t) \right\|_2^2 \tag{28}$$

Then we can write the L2 norm of the approximation series of $h$ as

$$\left\| \sum_{s=1}^{S} \langle h, g_s \rangle g_s(t) \right\|_2^2 = \sum_{s=1}^{S} \langle h, g_s \rangle^2 \tag{29}$$

due to the orthonormality of each pair of $\{g_s\}$. Therefore,

$$\|h(t)\|_2^2 - VF_S(\hat{f}_i, \mathbf{x}(t)) = \left\| \sum_{s=1}^{\infty} \langle h, g_s \rangle g_s(t) \right\|_2^2 - \left\| \sum_{s=1}^{S} \langle h, g_s \rangle g_s(t) \right\|_2^2$$

$$= \sum_{s=1}^{\infty} \langle h, g_s \rangle^2 - \sum_{s=1}^{S} \langle h, g_s \rangle^2 \tag{30}$$

$$= \sum_{s=S+1}^{\infty} \langle h, g_s \rangle^2$$

which is the sum of squared sine coefficients.

Suppose $l(t_0) = \lim_{t \to t_0} h(t)$. By Lemma 1 the coefficient converges as

$$\langle h, g_s \rangle = \sqrt{\frac{2}{T}} \left( \frac{T}{s\pi} \right) \left( (-1)^s l(T) - l(0) \right) + \sqrt{\frac{2}{T}} \left( \frac{T}{s\pi} \right)^2 O(\|h''(t)\|_2^2) \tag{31}$$

The dominating (first) term can be approximated using the power sum law. For $p > 1$,

$$\sum_{s=S+1}^{\infty} \frac{1}{s^p} = \frac{S^{1-p}}{p-1} + O\left( (S+1)^{-p} \right) \tag{32}$$

Therefore,

$$\|h(t)\|_2^2 - VF_S(\hat{f}_i, \mathbf{x}(t)) = O\left( \frac{1}{S} \right) \tag{33}$$

Furthermore, we can see that this upper bound cannot be improved since for some $\hat{f}_i$, $\left( (-1)^s l(T) - l(0) \right) \neq 0$. This implies the bound is indeed tight: $\|h(t)\|_2^2 - VF_S(\hat{f}_i, \mathbf{x}(t)) = \Theta\left( \frac{1}{S} \right)$. $\qquad \square$

**Lemma 1.** *Coefficient bound:*

$$\langle h, g_s \rangle = \sqrt{\frac{2}{T}} \frac{T}{s\pi} \left( (-1)^s l(T) - l(0) \right) + \sqrt{\frac{2}{T}} \left( \frac{T}{s\pi} \right)^2 \langle h'', g_s \rangle \tag{34}$$

*Proof.* Given $s$, consider coefficients

$$\langle h, g_s \rangle = \int_0^T h(t) g_s(t) \, dt \tag{35}$$

In this case, $g_s = \sqrt{\frac{2}{T}} \sin\left( \frac{s\pi t}{T} \right)$ is an element of the Fourier basis, and thus, the Fourier coefficients grow as follows:

$$\int_0^T h(t) g_s(t) \, dt = \int_0^T h(t) \sqrt{\frac{2}{T}} \sin\left( \frac{s\pi t}{T} \right) \, dt$$

$$= \sqrt{\frac{2}{T}} \left( \frac{T}{s\pi} h(t) \cos\left( \frac{s\pi t}{T} \right) \Big|_0^T - \frac{T}{s\pi} \int_0^T \cos\left( \frac{s\pi t}{T} \right) h'(t) \, dt \right)$$

$$= \sqrt{\frac{2}{T}} \frac{T}{s\pi} \left( (-1)^s l(T) - l(0) + \sin\left( \frac{s\pi t}{T} \right) h'(t) \, dt \Big|_0^T + \frac{T}{s\pi} \int_0^T \sin\left( \frac{s\pi t}{T} \right) h''(t) \, dt \right) \tag{36}$$

$$= \sqrt{\frac{2}{T}} \frac{T}{s\pi} \left( (-1)^s l(T) - l(0) \right) + \sqrt{\frac{2}{T}} \left( \frac{T}{s\pi} \right)^2 \left( \int_0^T h''(t) \sin\left( \frac{s\pi t}{T} \right) \, dt \right)$$

$$= \sqrt{\frac{2}{T}} \frac{T}{s\pi} \left( (-1)^s l(T) - l(0) \right) + \sqrt{\frac{2}{T}} \left( \frac{T}{s\pi} \right)^2 \langle h'', g_s \rangle.$$

This completes the proof. $\qquad \square$

## L  Appendix: Limitations

One limitation of our method is that it may not perform well on systems with high-dimensional ODEs, where larger state spaces exponentially expand the equation skeleton search space and amplify sensitivity to noise. This is a common challenge among existing studies. For future work, we plan to address this open research problem.

