# OpenReview forum: "Identifying Invariant Physical Dynamics Across Multiple Environments"
_TMLR — Under review for TMLR_

### Review · Reviewer_nBnA · 2026-02-27

**Summary Of Contributions:**

This paper considers the setting where multiple environments share a common unknown dynamical system structure but with different parameters. Given noisy, potentially imbalanced data from these environments, it proposes the IPAD framework to jointly discover the shared symbolic equation. IPAD leverages Monte Carlo Tree Search in the space of symbolic expressions, where at each iteration a skeleton equation is extracted and its constants are optimized separately per environment using Powell's method. A single-environment reward is computed based on a parsimony penalty and VF-based error. The individual rewards are aggregated into a multi-environment reward via a dataset-size-weighted mean and backpropagated through the MCTS tree. Once the search terminates, a post-hoc purification step computes each term's average significance per environment, removes insignificant terms, and selects the most frequently occurring purified term set across environments as the final skeleton.

The main novelty lies in:  (1) a novel algorithm combining MCTS with VF loss across environments, and (2) the reward design and purification method. (3) The prove that VF converges to L2 at rate $\mathcal{O}(1/S)$ where $S$ is the number of basis functions.

**Strength**:
- The algorithm is straightforward.
- The experiments coverage is thorough, as it covers both synthetic and real world problems with varying level of noise and balance of data across environments, the ablation study is well structured to demonstrate each blocks effect.

**Weakness**:
- The main blocks (MCTS approach and VF loss) are from existing researches, although the novel application is plausible, the 1. purification and 2. inclusion of complex penalty in reward definition seems quite practical. There is no theoretical analysis on whether the reward is actually consistent with the identifiability of the system.
- The experimental scale is limited to low-dimensional systems

**Audience:**

Yes

**Audience Explanation:**

The identifiability of dynamical systems across multiple similar environments is of both practical and research interest.

**Claims And Evidence:**

Yes

**Claims Explanation:**

The empirical results on the tested benchmarks are generally convincing, and the ablation studies provide useful evidence for each component's contribution. However, the experimental evidence would be strengthened by including more directly comparable baselines (e.g., SpReME (see suggested changes)), which addresses the same multi-environment setting.

**Requested Changes:**

- (Important) Please discuss and, if necessary, compare with SpReME (https://arxiv.org/pdf/2302.05942), which addresses similar multi-environment sparse ODE discovery problem.
- Figure 1 does not clearly distinguish the iterative MCTS loop (Steps 1–3) from the one-time post-hoc purification (Step 4), adding a stopping criterion block would be helpful for easier understanding.
- Please consider moving Table 1 outside the theorem environment for readability.
- Theorem 2 demonstrates the convergence rate of VF loss but does not address whether VF reward can guide MCTS toward structurally correct equations. It may be worth reconsidering its placement (e.g., moving to the appendix) or explain more directly how it is relevant to the IPAD framework.
- It will be helpful to provide empirical analysis of VF reward quality as a structural identification signal, for example, showing whether the top-ranked candidates by reward consistently have the correct skeleton would help clarify how the reward correlates with structural identifiability.

---

> ### Author Response · Authors · 2026-07-01
> **Response to Reviewer nBnA [Part 1/4]**
>
> We thank the reviewer for the insightful comments and suggestions.
>
> **Q1: The main blocks (MCTS approach and VF loss) are from existing researches, although the novel application is plausible, the 1. purification and 2. inclusion of complex penalty in reward definition seems quite practical. There is no theoretical analysis on whether the reward is actually consistent with the identifiability of the system.**
>
> **A1**: We thank the reviewer for this question. It raises two points: the novelty of our building blocks, and whether the reward is consistent with identifiability. We address each in turn, and for the second we discuss what can be proved theoretically for free-form symbolic regression.
>
> On novelty, our contribution lies not in the individual building blocks but in their coupling and in the setting to which they are applied: the VF-based data fit and parsimony reward, cross-environment aggregation, per-environment parameter optimization, MCTS search over free-form parse trees, and the post-hoc purification step are combined into a single procedure for recovering an invariant skeleton shared across environments.
>
> On whether the reward is consistent with identifiability, the formal question is whether the reward's maximizer over the grammar $\mathcal{G}$ of free-form parse trees coincides with the true minimal skeleton $f^*$.
>
> This reward combines the VF data-fit and parsimony terms (Eq. 7) with cross-environment aggregation (Eq. 8) and per-environment parameter optimization, and is maximized by MCTS. The actual procedure approximates an idealized one, and whether it recovers $f^*$ turns on three questions, of which only one is provable.
>
> (A) Effective MCTS search, given identifiability (assumed by necessity). Given a unique target, we further assume MCTS locates it rather than requiring an exact or exhaustive search, as it does empirically. In the worst case its sample complexity is at least exponential in tree depth: the UCT guarantees of Kocsis & Szepesvári (2006) are asymptotic, and Coquelin & Munos (2007) exhibit worst-case exponential regret.
>
> (B) Identifiability, given exact rewards (assumed by necessity). Indentifiability is provable only under stronger assumptions, for example within a fixed weak-form library (Zhang & Schaeffer, 2019; Brunton et al., 2016; Messenger & Bortz, 2021). For free-form search, it must be assumed because the maximizer is non-unique by construction (the same function admits many expressions, for example $x+x$ versus $2x$, or $\sin^2+\cos^2$ versus $1$) and the degree to which complexity is penalized is a data-dependent modeling choice with no objectively correct value.
>
> (C) How good finite rewards are compared to exact rewards, given optimal parameters (provable). The finite-$S$ VF reward converges to the idealized ground-truth MSE objective as $S \to \infty$ (Theorems 1 and 2), with an explicit rate. This is the one component that admits a proof.
>
> (D) Optimality of parameters (assumed by necessity). The reward is evaluated at per-environment parameters fitted by Powell's method, which we assume attains the global optimum. Because the candidate functions are free-form, we cannot use structural assumptions such as convexity or a restricted problem class, and global optimization of general nonconvex functions admits no universal guarantee. Even special cases such as nonconvex quadratic programming are NP-hard.
>
> Of these three, only the reward's fidelity to the idealized objective (C) is provable, and it is precisely Theorem 2; the other two are assumptions forced by the free-form setting. Theorem 2 is therefore the provable core of the reward–identifiability argument for our method, which is why we believe it belongs in the main text.  We acknowledge that our paper, like prior free-form symbolic regression & MCTS work, leaves these assumptions implicit rather than stating them, and we have made them explicit in the revised version.
>
> **Q2: The experimental scale is limited to low-dimensional systems.**
>
> As discussed in Appendix K, scaling to high-dimensional ODEs is a known limitation of IPAD and of symbolic regression methods more broadly. Based on our current findings, IPAD begins to struggle beyond approximately 8 dimensions, where larger state spaces exponentially expand the equation skeleton search space and amplify sensitivity to noise. We have expanded this discussion in the revised version (Appendix L). Extending IPAD to high-dimensional dynamical systems remains an important open problem that we plan to pursue in future work.

---

> ### Author Response · Authors · 2026-07-01
> **Response to Reviewer nBnA [Part 2/4]**
>
> **Q3: (Important) Please discuss and, if necessary, compare with SpReME (https://arxiv.org/pdf/2302.05942), which addresses similar multi-environment sparse ODE discovery problem.**
>
> **A3**: Thank you for pointing out this related work.
>
> (i) SpReME does not provide source code in their arXiv paper, which prevents direct experimental comparison. Nonetheless, a key conceptual distinction is worth noting: SpReME requires a pre-specified candidate function library, including prior domain knowledge (e.g., "... for the system of the damped pendulum, we additionally use trigonometric feature function sin(x1(t)), sin(x2(t)) and sin(x1(t) + x2(t)) as candidate") for pendulum dynamics) - which is reflected in their own framing that SpReME "uncovers governing dynamics from multiple environments with the help of incomplete prior knowledge." IPAD, by contrast, applie a tree search policy powered by MCTS and requires no such prior knowledge or hand-crafted candidate library, making it applicable in settings where the functional form of the dynamics is unknown.
>
> (ii) Although a direct comparison with SpReME is not feasible due to the absence of publicly available code, we note that SpReME uses SINDy as a baseline. We have included SINDy in our own comparisons, along with an additional baseline PySR.
>
> (iia) Compared to SINDy:
> The dataset settings used are Setting 1 and Setting 2 in our submission. For simplicity and consistency, we report results only from environment 1. The experiments are conducted under three noise levels: $σ_R$ = 0.00, 0.10, and 0.20. We focus on the Lotka-Volterra system, specifically Equation 1: $dx_0/dt = \alpha x_0 − \beta x_0 x_1$. The discovery results using SINDy are summarized in Table T2.
>
> We can see that SINDy's performance is highly sensitive to the noise level. Under both Setting 1 and Setting 2, SINDy successfully recovered the correct equation only when σ_R = 0.00. In contrast, as shown in Table 2 of the main paper, IPAD achieves a significantly higher success rate (0.95–1.00) across the same dataset configurations.
>
> Table T2: SINDy discovery results for Equation 1 of the Lotka-Volterra system.
> |Setting|σ_R = 0.00|σ_R = 0.10|σ_R = 0.20|
> |--|--|--|--|
> |Setting1|**1.0 x0 + -0.30 x0 x1**|-0.144 1 + 0.988 x0 + -0.406 x1 + -0.261 x0 x1|-1.207 1 + 0.991 x0 + 0.101 x1 + -0.212 x0 x1 + -0.137 x1^2|
> |Setting2|**1.0 x0 + -0.30 x0 x1**|-0.734 1 + 1.024 x0 + 1.274 x1 + -0.352 x0 x1 + -0.137 x1^2|  -1.866 1 + 1.033 x0 + 3.218 x1 + -0.392 x0 x1 + -0.387 x1^2|
> |Success|No|No|No|
>
> (iib) Compared to PySR
>
> The dataset configurations used are the same as Table T2. The reported score from PySR is used to rank candidate equations. The summary of discovery performance is presented in Table T3, and an example output from PySR for Setting 2 with $σ_R$ = 0.10 is shown in Table R5.
>
> Our results indicate that PySR's performance is also highly sensitive to noise. Under both Setting 1 and Setting 2, PySR successfully discovers the correct equation only when $σ_R$ = 0.00. In contrast, as shown in Table 2 of the main paper, IPAD maintains a much higher success rate (0.95–1.00) under the same configurations.
>
> Table T3: PySR discovery results for Equation 1 of the Lotka-Volterra system.
>
> |Setting|σ_R = 0.00|σ_R = 0.10|σ_R = 0.20|
> |--|--|--|--|
> |Setting1|**x0 - (x0 * (x1 * 0.29987276))**|x0 - x1|x0 - x1|
> ||score: 9.164871|score: 0.019424|score: 0.004863|
> |Setting2|**((x1 * -0.29979765) + 0.9997918) * x0**|x0 / exp(x1)|x0 / exp(x1)|
> ||score: 9.68168|score: 0.008668|score: 0.002037|
> |Success|**Yes**|No|No|
>
> [1] (SINDy) Brunton, Steven L., Joshua L. Proctor, and J. Nathan Kutz. "Discovering governing equations from data by sparse identification of nonlinear dynamical systems." Proceedings of the national academy of sciences 113.15 (2016): 3932-3937.
>
> [2] (PySR) Cranmer, Miles. "Interpretable machine learning for science with PySR and SymbolicRegression. jl." arXiv preprint arXiv:2305.01582 (2023).
>
> **Q4: Figure 1 does not clearly distinguish the iterative MCTS loop (Steps 1–3) from the one-time post-hoc purification (Step 4), adding a stopping criterion block would be helpful for easier understanding.**
>
> **A4**: We have revised Figure 1 in the revised version to add an explicit stopping condition with a detailed highlighted criterion for it in the main text (Section 4.3) to make the loop structure easier to follow.
>
> **Q5: Please consider moving Table 1 outside the theorem environment for readability.**
>
> **A5**: We have moved Table 1 outside the theorem environment (move to page 9) in the revised version to improve readability.
>
> **Q6: Theorem 2 demonstrates the convergence rate of VF loss but does not address whether VF reward can guide MCTS toward structurally correct equations. It may be worth reconsidering its placement (e.g., moving to the appendix) or explain more directly how it is relevant to the IPAD framework.**
>
> **A6**: As discussed in A1.

---

> ### Author Response · Authors · 2026-07-01
> **Response to Reviewer nBnA [Part 3/4]**
>
> **Q7: It will be helpful to provide empirical analysis of VF reward quality as a structural identification signal, for example, showing whether the top-ranked candidates by reward consistently have the correct skeleton would help clarify how the reward correlates with structural identifiability.**
>
> **A7**: Thank you for this suggestion. We provide the requested empirical analysis in two parts: (i) an aggregate analysis over all runs, quantifying how often the highest-reward candidate carries the correct skeleton; and (ii) representative per-run leaderboards illustrating the ranking in each regime. Throughout, candidates are ranked by the raw VF reward `r_ME`, skeletons are shown with their coefficients written as `C`, and ranking is performed within each run only, since `r_ME` is not comparable across runs or noise levels.
>
> We evaluate on a grid of four simulated systems and their ODE variables, under noise ratios $\{0.00, 0.10, 0.20\}$ in the balanced setting, with the full per-configuration budget and 10 seeds per cell (30 cells × 10 seeds = 300 runs).
>
> **Table T4: Rank of the correct skeleton under VF-reward ranking, by system.** Each count is the number of runs ($n$) in which the correct skeleton, after the standard purification step, appears at rank 1, within the top 5, outside the top 5, or is absent from the candidate set.
>
> | System | n | in top-1 | in top-5 | outside top-5 | absent |
> |---|---|---|---|---|---|
> | Lotka–Volterra | 60 | 60 | 60 | 0 | 0 |
> | Lorenz | 90 | 69 | 69 | 17 | 4 |
> | SIR | 90 | 90 | 90 | 0 | 0 |
> | Damped Pendulum | 60 | 59 | 60 | 0 | 0 |
> | **TOTAL** | **300** | **278 (92.7%)** | **279** | **17** | **4** |
>
> Ranking by VF reward places the correct skeleton (after the standard purification step) at rank 1 in 93% of all runs and in 94% of the runs in which the correct structure is reachable. The only residual weakness is the stiff Lorenz cross-term cell, and outright misses are rare (4 of 300). The top reward is therefore a consistent structural-identification signal. We illustrate the typical ranking with three representative runs below.
>
> Example 1:
> **Table T5: Lotka–Volterra, ODE 1, noise 0.00 (truth `1.0·x − 0.3·x·y`); a decisive win.**
>
> | # | r_ME | raw skeleton | purified skeleton | correct? |
> |---|---|---|---|---|
> | 1 | **0.99686** | `−C·x·y + C·x` | `−C·x·y + C·x` | Y |
> | 2 | 0.00021 | `−C·x·y + C·x + C·y²` | `−C·x·y + C·x + C·y²` | N |
> | 3 | 0.00018 | `−C·x·y²` | `−C·x·y²` | N |
> | 4 | 0.00017 | `−C·x·y² − C·x` | `−C·x·y² − C·x` | N |
> | 5 | 0.00014 | `C·x − C·y` | `C·x − C·y` | N |
> | 6 | 0.00014 | `−C·y + C` | `−C·y + C` | N |
> | 7 | 0.00014 | `−C·x·y − C·y²` | `−C·x·y − C·y²` | N |
> | 8 | 0.00014 | `−C·x·y³` | `−C·x·y³` | N |
> | 9 | 0.00013 | `−C·y² + C·y` | `−C·y² + C·y` | N |
> | 10 | 0.00013 | `−C·y²` | `−C·y²` | N |
>
> Here 974 candidates were evaluated over 106 distinct structures, with `r_ME ∈ [0, 0.997]`. The rank-1 reward of 0.997 exceeds the rank-2 reward of 0.0002 by a margin of 0.997, and the rank-1 fitted equation (environment 0) is `−0.3·x·y + 1.0·x`.
>
> ---
>
> Example 2:
>
> **Table T6: SIR, ODE 1, noise 0.00 (truth `−0.01·x·y`); a clean win, with reward concentrated on the structural neighborhood of the truth.**
>
> | # | r_ME | raw skeleton | purified skeleton | correct? |
> |---|---|---|---|---|
> | 1 | **0.99945** | `−C·x·y` | `−C·x·y` | Y |
> | 2 | 0.99912 | `−C·x·y − C·y²` | `−C·x·y` | N → Y |
> | 3 | 0.99761 | `−C·x² − C·x·y` | `−C·x·y` | N → Y |
> | 4 | 0.99595 | `C·x² − C·x·y` | `−C·x·y` | N → Y |
> | 5 | 0.99124 | `−C·x·y + C·y·z` | `−C·x·y` | N → Y |
> | 6 | 0.98747 | `−C·x·y − C·x·z` | `−C·x·y` | N → Y |
> | 7 | 0.97378 | `−C·x·y − C·x` | `−C·x·y` | N → Y |
> | 8 | 0.57649 | `−C·x·y + C·x·z` | `−C·x·y` | N → Y |
> | 9 | 0.21611 | `C·x² − C·x` | `−C·x` | N |
> | 10 | 0.13505 | `−C·x·y + C·x` | `−C·x·y + C·x` | N |
>
> Here 2672 candidates were evaluated over 212 distinct structures, with `r_ME ∈ [0, 0.999]`. The rank-1-to-rank-2 margin is only 0.0003, but ranks 1 through 8 all sit at `r > 0.57` before the reward drops sharply to 0.22 at rank 9. Each of these eight is the truth `−C·x·y` plus or minus one spurious term, and all of them purify back to `−C·x·y`. The rank-1 fitted equation (environment 0) is `−0.01·x·y`.

---

> ### Author Response · Authors · 2026-07-01
> **Response to Reviewer nBnA [Part 4/4]**
>
> **A7 (Continued)**:
>
> Example 3 (Negative Case):
>
> **Table T7: Lorenz, ODE 2, noise 0.00 (truth `28·x − x·z − y`); the difficult, truth-absent (negative) case (Please Note that this occurs rarely in practice, as reflected in the main result tables; we highlight this case here purely for illustrative purposes.).**
>
> | # | r_ME | raw skeleton | purified skeleton | correct? |
> |---|---|---|---|---|
> | 1 | **0.28776** | `−C·x·z + C·x` | `−C·x·z + C·x` | N |
> | 2 | 0.05994 | `−C·z² + C·z` | `−C·z² + C·z` | N |
> | 3 | 0.01632 | `−C·x·z + C·x + C·y` | `−C·x·z + C·x + C·y` | N |
> | 4 | 0.01118 | `−C·x·z + C·y + C·z` | `−C·x·z + C·y + C·z` | N |
> | 5 | 0.01067 | `C·y − C·z²` | `C·y − C·z²` | N |
> | 6 | 0.01012 | `C·x − C·z²` | `C·x − C·z²` | N |
> | 7 | 0.00830 | `C·x − C·y·z²` | `C·x − C·y·z²` | N |
> | 8 | 0.00724 | `−C·x²·z²` | `−C·x²·z²` | N |
> | 9 | 0.00707 | `−C·x² + C·y` | `−C·x² + C·y` | N |
> | 10 | 0.00694 | `−C·x·y·z + C·y` | `−C·x·y·z + C·y` | N |
>
> Here 1887 candidates were evaluated over 264 distinct structures, with `r_ME ∈ [0, 0.288]`. The rank-1-to-rank-2 margin is wide (0.228), yet the rank-1 candidate is incorrect: the fitted equation (environment 0) is `−0.951·x·z + 25.548·x`, with the `−y` term missing. In this case the three-term truth was never generated as a candidate.

---

### Review · Reviewer_FVxD · 2026-02-27

**Summary Of Contributions:**

This paper presents a new method to identify the physical laws from from multiple datasets that
have been collected across multiple datasets. The output of the method is in the end an
interpratable symbolic equation.

Overall their method consists of x stages.
1) The "skeleton" equations are extracted by using MCTS.
2) The hyperparameters of the "skeleton" equations are optimized.
3) Purification/Prunning to reduce the complexity of the outcome.

The method is overall novel and of potential interest for the TMLR community.

**Audience:**

Yes

**Audience Explanation:**

Finding the coverning equations of system is a common problem found in many engineering problems, e.g. dynamical systems modeling.

**Claims And Evidence:**

Yes

**Claims Explanation:**

The authors support their claims via detailed theoretical analysis as well as experiments.

**Requested Changes:**

While the authors acknowledge the limitations of their method in a high dimensional space, I think this
discussion could still be improved. For example, does your method also apply to a robot dynamics? Can you do
experiments where you check for the breaking point?

The authors measure the performance of baseline methods via "success rate". Maybe providing also
other metrics such as MSE can be beneficial, as "success rate" is binary.

---

> ### Author Response · Authors · 2026-07-01
> **Response to Reviewer FVxD**
>
> We thank the reviewer for the insightful comments and suggestions.
>
> **Q1: While the authors acknowledge the limitations of their method in a high dimensional space, I think this discussion could still be improved. For example, does your method also apply to a robot dynamics? Can you do experiments where you check for the breaking point?.**
>
> **A1**: Thank you for this suggestion. As discussed in Appendix K, handling high-dimensional ODEs is a known limitation of IPAD and of symbolic regression methods more broadly. Robot dynamics represent a canonical example of this challenge - manipulators (e.g., a 6-DOF arm) operate in 12-dimensional state spaces, quadrupeds (e.g., Spot) in 24–36 dimensions, and humanoids (e.g., Atlas) in 50–100+ dimensions. Based on our current findings, IPAD begins to struggle beyond approximately 8 dimensions, as larger state spaces exponentially expand the equation skeleton search space and amplify sensitivity to noise. Extending IPAD to high-dimensional dynamical systems remains an important open problem that we plan to pursue in future work.
>
>
> **Q2: The authors measure the performance of baseline methods via "success rate". Maybe providing also other metrics such as MSE can be beneficial, as "success rate" is binary.?**
>
> **A2**: Thank you for this suggestion. We also recorded error-based metrics such as MSE early in our work, but found that lower MSE does not reliably indicate better results for our research objective. We illustrate this with a concrete comparison against MetaPhysica (ICLR 2024 [1]), described in Section 5 and detailed in Appendix F.
> The results are presented in Table T1 below. Although MetaPhysica achieves lower MSE, IPAD achieves a substantially higher discovery rate - because MetaPhysica fails to explicitly recover the true invariant physical laws. The core reason is that MetaPhysica operates as a regression approximation, fitting an overly complex equation to match even a simple ground-truth expression, and the resulting surrogate may not faithfully represent the underlying dynamics.
>
> **Table T1: Comparison of success rate(↑) between IPAD and MetaPhysica.** The experiments are conducted on the Lotka-Volterra system using 20 random seeds  and 2 equations under Dataset Distribution Setting 1 (each environment has 500 data points).
> |    Setting    | Noise Ratio   |     IPAD(Ours)| MetaPhysica  |
> |-|-|-|-|
> |    Setting 1  |     0.00      |     0.98  |     0.00     |
> |    Setting 1  |     0.05      |     0.95  |     0.00     |
> |    Setting 1  |     0.10      |     0.98  |     0.00     |
> |    Setting 1  |     0.15      |     1.00  |     0.00     |
> |    Setting 1  |     0.20      |     0.98  |     0.00     |
>
> For instance, with noise ratio 0.00, the ground truth for equation 1 is $\dfrac{dx_0}{dt}=C_1\cdot x_0+C_2\cdot x_0\cdot x_1$ with $C_{1,2}=\{-0.39, 1.2\}$. However, MetaPhysica returns $\dfrac{dx_0}{dt}=C_1\cdot x_0^3 - C_2\cdot x_0^2\cdot x_1 + C_3\cdot x_0^2 - C_4\cdot x_0\cdot x_1^2 - C_5\cdot x_0\cdot x_1 + C_6\cdot x_0 + C_7\cdot x_1^3$ $- C_8\cdot x_1^2 - C_9\cdot x_1 + C_{10}\cdot sin(x_0) + C_{11}\cdot sin(x_1) - C_{12}\cdot cos(x_0) + C_{13}\cdot cos(x_1) + C_{14}$, with constants $C_{1..14}=\{-0.000191, 0.001667, 0.019217, -0.013490, -0.157484, 0.616406, 0.016312$
> $-0.224740, -0.210307, 0.026757, 0.705423, -0.175709, 0.745682, 1.461279\}$ - a heavily over-parameterized surrogate that happens to fit the data but bears no resemblance to the true dynamics. In contrast, IPAD recovers the correct equation skeleton and outputs $C_{1,2}=\{-0.389936, 1.199808\}$, closely matching the ground truth.
>
> [1] MetaPhysiCa: OOD Robustness in Physics-informed Machine Learning. ICLR, 2024. https://arxiv.org/pdf/2303.03181

---

> > ### Comment · Reviewer_FVxD · 2026-07-03
> > **Comment**
> >
> > Thank you for the  details. However, I still think the manuscript has not a sufficient discussion regarding the MSE. The tables are still driven by the success rate.

---

### Review · Reviewer_78NP · 2026-06-21

**Summary Of Contributions:**

This paper proposes IPAD, a symbolic regression framework that identifies a shared ODE structure across multiple environments while estimating environment-specific coefficients. The method combines MCTS, a variational-formulation loss, multi-environment aggregation, and post-hoc purification. The topic is relevant, and the empirical results are promising.

Strengths:

- The problem is relevant to symbolic regression and scientific machine learning.
- A shared skeleton with environment-specific coefficients is interpretable and practically useful.
- The experiments cover noise, sample imbalance, several systems, and component ablations.

Main weaknesses:

- The VF objective and theorem are not stated consistently.
- The experiments do not test the claimed adaptation to unseen environments.
- The baseline aggregation is ground-truth-dependent and is not a fair unified-model comparison.

**Audience:**

Yes

**Audience Explanation:**

Yes.
Researchers in symbolic regression, system identification, weak-form learning, and multi-environment modeling would be interested in recovering a common interpretable equation structure from heterogeneous noisy trajectories. The method and initial results are relevant, even though the evidence and presentation require substantial correction.

**Broader Impact Concerns:**

The method itself raises no major ethical concern. The COVID-19 result could, however, be misinterpreted as a causal or policy-relevant epidemiological model. The paper should state that it is not validated for forecasting or intervention design and briefly discuss reporting bias, smoothing, and time-varying disease dynamics.

**Claims And Evidence:**

No

**Claims Explanation:**

Not yet. Three major issues affect the main conclusions.

1. The VF formulation is internally inconsistent. Equation (2) differs from Equations (9) and (20) and from the Parseval-based proof. The theorem also contains an impossible assumption, $|h''|_2<0$, and apparent derivation errors. Since VF is central to the method, the authors must provide a consistent objective, correct the theorem and proof, define aggregation over multiple trajectories, and clarify which implementation produced the results.

2. The baseline comparison does not fairly evaluate unified equation recovery. IPAD jointly uses all environments, whereas SPL and D-code are applied separately. Appendix D aggregates ground-truth correctness indicators rather than combining predicted equations into one skeleton. The success-rate unit and symbolic-equivalence criterion are also unclear. The paper needs a ground-truth-independent aggregation rule and a credible pooled or shared-support baseline.

3. Unseen-environment adaptation is not evaluated. All environments are used during structure discovery. The experiments therefore support recovery across observed environments, but not adaptation or generalization to unseen environments. The authors should add held-out-environment experiments or restrict the corresponding claims.

**Requested Changes:**

### Critical to Acceptance

1. **Correct and consistently define the VF formulation.** The objective, theorem, proof, and implementation must use the same mathematical definition. The authors should also clarify how VF losses are aggregated across multiple trajectories and rerun experiments if the implementation differs from the corrected formulation.

2. **Redesign the baseline comparison and clarify the evaluation protocol.** The current aggregation uses ground-truth correctness rather than producing a unified predicted equation, making the comparison with IPAD inequivalent. The authors should introduce a ground-truth-independent aggregation method or a pooled/shared-support baseline and clearly define the success unit and symbolic-equivalence criterion.

3. **Evaluate unseen-environment adaptation or narrow the claims.** Since all evaluated environments are used for structure discovery, the experiments do not support adaptation or generalization to unseen environments. The authors should add held-out-environment experiments or restrict the claims to shared-structure recovery across observed environments.

### Changes That Would Strengthen the Work

1. **Separate hyperparameter selection from final evaluation.** Parameters such as $S$ and $\tau$ should be selected using validation data or prespecified independently of the reported test results to reduce potential tuning bias.

2. **Reframe and strengthen the real-world evaluation.** The COVID-19 SIR equation should be described as an assumed reference model rather than ground truth. The pendulum experiment should clarify time rescaling and damping assumptions and report recovered coefficients and predictive errors.

---

> ### Author Response · Authors · 2026-07-01
> **Response to Reviewer 78NP [Part 1/2]**
>
> We thank the reviewer for the insightful comments and suggestions.
>
> **Q1: The VF formulation is internally inconsistent. Equation (2) differs from Equations (9) and (20) and from the Parseval-based proof. The theorem also contains an impossible assumption, |h''|2<0, and apparent derivation errors. Since VF is central to the method, the authors must provide a consistent objective, correct the theorem and proof, define aggregation over multiple trajectories, and clarify which implementation produced the results.**
>
> **A1**: Thank you for catching these oversights. About aggregation over multiple trajectories, for each environment, we find the uniform mean $r_k = \frac{1}{M_k}\sum_{m=1}^{M_k} r_k^{(m)}$, where $M_k$ is the number of trajectories observed in environment $k$ (each with $|D_k|$ time samples), and $r_k^{(m)}$ is the single-environment reward of Eq.~(7) on the $m$-th trajectory. Environments are then combined by the dataset-size–weighted mean defined in Eq. (8) $r_{ME}=\sum_k |D_k| r_k / \sum_k |D_k|$.
>
> As for other mentioned issues, they are typos to be corrected below. None affects the implementation or results, as the released code already uses the correct objective.
> - Opening parenthesis "(" sits before $\sum$ for equation (2), (9) and (20): $$\text{VF}_S(\hat f_i,x) = \sum_{s=1}^S \left(\int_0^T \hat f_i(x(t)) g_s(t) dt + \int_0^T x_i(t) \dot g_s(t) dt\right)^2.$$
> - Theorem 1 limit: argument $f_i \to \hat f_i$, i.e. $\lim_{S\to\infty} VF_S(\hat f_i,x) = \|\hat f_i(x) - f_i(x)\|_2^2$.
> - Convergence-rate assumption: $\|h''\|_2 < 0 \to \|h''\|_2 < \infty$.
>
> We have corrected all three in the revised version.
>
>
> **Q2: The baseline comparison does not fairly evaluate unified equation recovery. IPAD jointly uses all environments, whereas SPL and D-code are applied separately. Appendix D aggregates ground-truth correctness indicators rather than combining predicted equations into one skeleton. The success-rate unit and symbolic-equivalence criterion are also unclear. The paper needs a ground-truth-independent aggregation rule and a credible pooled or shared-support baseline.**
>
> **A2**: Thank you for raising this point. We address each concern in turn.
>
> - **Shared-support baseline.** We do include a pooled baseline, MetaPhysica, described in Section 5 and detailed in Appendix F. To our knowledge, MetaPhysica is the only existing method capable of leveraging data from all environments simultaneously to recover a shared hidden ODE system; all other baselines operate on a single environment at a time. However, MetaPhysica discovers surrogate equations rather than true governing equations, which results in a 0% success rate in our experiments - a limitation documented in Section 5 and Appendix F.
> - **Ground-truth-independent aggregation.** To aggregate results from single-environment baselines as fairly as possible, we employ two aggregation strategies: Average (Avg) and Weighted Vote (W.V.), introduced in Section 5 and detailed in Appendix D. Both are ground-truth-independent by construction, and their results are reported in Tables 2–3 (Section 5) and Tables 14–15 (Appendix G).
>
> We hope this clarifies the comparison methodology and the fairness of our evaluation.
>
> **Q3: Unseen-environment adaptation is not evaluated. All environments are used during structure discovery. The experiments therefore support recovery across observed environments, but not adaptation or generalization to unseen environments. The authors should add held-out-environment experiments or restrict the corresponding claims.**
>
> **A3**: Our work addresses a setting where data from multiple environments (e.g., fox–rabbit dynamics in Forest A, wolf–sheep in Forest B, tiger–deer in Forest C) are assumed to share an underlying differential equation structure. The goal is to recover this hidden pattern from observed data when the governing equations are unknown.
> Crucially, this framework naturally extends to unseen environments. Given a new dataset from an unobserved environment (e.g., a novel predator–prey system in Forest D), the equation skeleton learned from Forests A-C remains directly applicable. Adapting to Forest D then reduces to a simple parameter optimization step - a well-established technique - which mirrors exactly what is already performed for each training environment (Step 2 of Figure 1). In other words, generalizing to a new environment requires only one additional parameter optimization pass, without any modification to the structure discovery phase.
> We therefore respectfully clarify that IPAD inherently possesses generalization capability - the learned equation skeleton serves as the transferable component across environments, and adapting it to any unseen setting reduces to a straightforward parameter optimization by design. We have added a discussion section (detailed in the new Appendix J and referenced in Section 5.1) to the revised version to make this capability explicit.

---

> ### Author Response · Authors · 2026-07-01
> **Response to Reviewer 78NP [Part 2/2]**
>
> **Q4: Separate hyperparameter selection from final evaluation. Parameters such as S and tau should be selected using validation data or prespecified independently of the reported test results to reduce potential tuning bias.**
>
> **A4**: We would like to clarify that the hyperparameters S and τ were selected prior to running any of the main experiments. Their selection is documented in Table 17 (Appendix I) and Figure 3 (Section 5.2), and all reported results - including Tables 2–3 (Section 5) and Tables 14–15 (Appendix G) - use these fixed hyperparameters throughout. We have also clarified this (update in Appendix I) in our revised version.
>
> **Q5: Reframe and strengthen the real-world evaluation. The COVID-19 SIR equation should be described as an assumed reference model rather than ground truth. The pendulum experiment should clarify time rescaling and damping assumptions and report recovered coefficients and predictive errors.**
>
> **A5**: Thank you for these suggestions.
>
> - **COVID-19 SIR equation.** We agree with this framing and have relabeled the SIR equation as an "assumed reference model" rather than "ground truth" in the revised version.
> - **Pendulum experiment.** The full configuration and experimental settings are detailed in Appendix C.6. Time is recorded directly in seconds with no rescaling applied. The recovered coefficients and predictive errors are reported in the first part of Appendix H.
>
> **Q6: The method itself raises no major ethical concern. The COVID-19 result could, however, be misinterpreted as a causal or policy-relevant epidemiological model. The paper should state that it is not validated for forecasting or intervention design and briefly discuss reporting bias, smoothing, and time-varying disease dynamics.**
>
> **A6**: We fully agree that the COVID-19 result should not be interpreted as a validated epidemiological model for forecasting or intervention design. We have included an ethics discussion in the revised version (Appendix C.5) explicitly stating this.

---

### Comment · Action_Editor_XXVU · 2026-06-22
**Start of discussion phase**

Dear authors, dear reviewers,

First, apologies for the severe delay on this submission. I was chasing different sets of reviewers for quite some time until we finally got in the three reviews.
@Reviewers: Thanks a lot for your time and submitting the reviews.
@Authors: Please carefully consider the comments/suggestions/questions in the reviews and start the discussion phase.

The discussion phase will last 2 weeks. After 2 weeks, reviewers will be able to submit their formal decision recommendation.

Best,
Action Editor

---

### Comment · Action_Editor_XXVU · 2026-07-07
**Please engage in discussion**

Dear reviewers,

The authors have now provided extensive replies to your reviews. Please read through the rebuttal carefully and consider how it affects your assessment of the submission. Please engage in a discussion with the authors.
Once you feel you have all the information you need, you are now also able to submit an "official recommendation".
Once all reviewers have submitted the "official recommendation" we will be able to proceed with the submission.

Best,
AE